# Single-cell transcriptomics identifies the differentiation trajectory from inflammatory monocytes to pro-resolving macrophages in a mouse skin allergy model

Kensuke Miyake [1] ✉, Junya Ito[1], Kazufusa Takahashi[1], Jun Nakabayashi[2], Frank Brombacher[3], Shigeyuki Shichino [4], Soichiro Yoshikawa[5], Sachiko Miyake[5] & Hajime Karasuyama [1]

Both monocytes and macrophages are heterogeneous populations. It was traditionally understood that Ly6C$^{hi}$ classical (inflammatory) monocytes differentiate into pro-inflammatory Ly6C$^{hi}$ macrophages. Accumulating evidence has suggested that Ly6C$^{hi}$ classical monocytes can also differentiate into Ly6C$^{lo}$ pro-resolving macrophages under certain conditions, while their differentiation trajectory remains to be fully elucidated. The present study with scRNA-seq and flow cytometric analyses reveals that Ly6C$^{hi}$PD-L2$^{lo}$ classical monocytes recruited to the allergic skin lesion sequentially differentiate into Ly6C$^{lo}$PD-L2$^{hi}$ pro-resolving macrophages, via intermediate Ly6C$^{hi}$PD-L2$^{hi}$ macrophages but not Ly6C$^{lo}$ non-classical monocytes, in an IL-4 receptor-dependent manner. Along the differentiation, classical monocyte-derived macrophages display anti-inflammatory signatures followed by metabolic rewiring concordant with their ability to phagocytose apoptotic neutrophils and allergens, therefore contributing to the resolution of inflammation. The failure in the generation of these pro-resolving macrophages drives the IL-1α-mediated cycle of inflammation with abscess-like accumulation of necrotic neutrophils. Thus, we clarify the stepwise differentiation trajectory from Ly6C$^{hi}$ classical monocytes toward Ly6C$^{lo}$ pro-resolving macrophages that restrain neutrophilic aggravation of skin allergic inflammation.

Monocytes are blood-circulating mononuclear cells which maintain tissue homeostasis and contribute to various immune responses. They are composed of heterogeneous subsets in both humans and mice: classical monocytes (comprising 80–90 % of the monocyte pool) and non-classical monocytes[1,2]. In mice, these two subsets can be distinguished by the differential expression of Ly6C, a member of the GPI-anchored Ly6 family, on the cell surface. Under homeostatic conditions, Ly6C$^{hi}$ classical monocytes are generated from common monocyte progenitors in the bone marrow, and a part of them further differentiate into Ly6C$^{lo}$ non-classical monocytes in the peripheral

[1]Inflammation, Infection and Immunity Laboratory, Advanced Research Institute, Tokyo Medical and Dental University (TMDU), Tokyo, Japan. [2]College of Liberal Arts and Sciences, Tokyo Medical and Dental University (TMDU), Tokyo, Japan. [3]Institute of Infectious Disease and Molecular Medicine, International Center for Genetic and Biotechnology Cape Town Component & University of Cape Town, Cape Town, South Africa. [4]Division of Molecular Regulation of Inflammatory and Immune Diseases, Research Institute of Biomedical Sciences, Tokyo University of Science, Noda, Japan. [5]Department of Immunology, Juntendo University School of Medicine, Tokyo, Japan. ✉e-mail: miyake.mbch@tmd.ac.jp

blood. After the egress from the bone marrow, some of monocytes enter peripheral tissues and differentiate into tissue-resident-like macrophages to replenish the infection/tissue damage-induced loss of genuine tissue-resident macrophages with embryonic origin[3]. Under inflammatory conditions, monocytes are recruited to the site of inflammation and rapidly differentiate into macrophages which display functional and transcriptional properties distinct from those of tissue-resident macrophages[2,4–6]. Traditionally, Ly6C[hi] classical monocytes have been shown to differentiate to Ly6C[hi] macrophages with pro-inflammatory properties whereas Ly6C[lo] non-classical monocytes differentiate into Ly6C[lo] macrophages with anti-inflammatory and pro-resolving functions. Accordingly, classical monocytes used to be called "inflammatory" monocytes.

We now appreciate that the developmental relationship between each subset of monocytes and that of macrophages is more plastic[2,4,5]. Ly6C[hi] classical monocytes can also differentiate into Ly6C[lo] macrophages, facilitating the resolution of inflammation and tissue repair, rather than promoting inflammation, in murine models of myocardial infarction, atherosclerosis and acute injury of the liver, muscle and spinal cord[7–14]. It has been proposed that Ly6C[hi] classical monocytes firstly differentiate to Ly6C[lo] non-classical monocytes and subsequently into anti-inflammatory Ly6C[lo] macrophages[15,16]. Other studies demonstrated that Ly6C[lo] macrophages can be generated from Ly6C[hi] classical monocytes even in the absence of blood circulating Ly6C[lo] non-classical monocytes[17,18]. It was also suggested that Ly6C[lo] macrophages can be differentiated from Ly6C[hi] macrophages[9,19]. Thus, the differentiation trajectory of anti-inflammatory Ly6C[lo] macrophages remains less clear compared to that of pro-inflammatory Ly6C[hi] macrophages.

We previously reported in a murine model of skin allergic inflammation (IgE-mediated chronic allergic inflammation, IgE-CAI) that skin-recruited Ly6C[hi] classical monocytes differentiate into anti-inflammatory macrophages, leading to the termination of allergic inflammation in the skin[20]. Ly6C[hi] classical monocytes express high levels of a chemokine receptor CCR2 on the cell surface. Therefore, in CCR2-deficient ($Ccr2^{-/-}$) mice, few or no classical monocytes are recruited to the skin lesion, resulting in poor generation of anti-inflammatory macrophages and exaggerated skin inflammation.

In this study, we use this IgE-CAI model and combined single-cell RNA-seq (scRNA-seq), flow cytometric and functional analyses to explore the differentiation trajectory from classical monocytes to anti-inflammatory macrophages in the skin lesion and to clarify mechanisms by which monocyte-derived macrophages exert anti-inflammatory functions.

## Results
### scRNA-seq analysis identifies four distinct clusters of monocyte-macrophage lineage cells (Mo-Macs) in the IgE-CAI skin lesion

$Ccr2^{-/-}$ mice showed exaggerated and prolonged ear swelling with severer erythema and increased cellular accumulation in the IgE-CAI skin lesion when compared to wild-type (WT) mice (Fig. 1a–c). Of note, the number of neutrophils highly increased in $Ccr2^{-/-}$ mice whereas that of macrophages diminished in $Ccr2^{-/-}$ mice (Fig. 1c). Adoptive transfer of CD115[+] monocytes isolated from WT mice into $Ccr2^{-/-}$ mice dampened exaggerated ear swelling and cell accumulation in the skin lesion (Fig. 1d, e), suggesting that macrophages derived from CCR2[+] monocytes contributed to the resolution of skin inflammation in WT mice.

To explore the trajectory of differentiation from monocytes to pro-resolving macrophages in the skin lesion and to clarify mechanisms by which monocyte-derived macrophages dampen excess inflammation, we conducted scRNA-seq analysis of cells isolated from IgE-CAI skin lesions of WT and $Ccr2^{-/-}$ mice on day 3 and 5 post-allergen challenge. The visualization of scRNA-seq data with dimensionality reduction using uniform manifold approximation and projection

(UMAP) identified a broad spectrum of hematopoietic and non-hematopoietic cell lineages (Supplementary Fig. 1a). In accordance with the flow cytometric analysis (Fig. 1c), $Ccr2^{-/-}$ mice showed decreased accumulation of Mo-Macs while increased neutrophil accumulation (Supplementary Fig. 1b).

Re-clustering of Mo-Mac and DC populations (clusters 3, 9, 12, 13, 20 in Supplementary Fig. 1a) identified 6 new clusters (Supplementary Fig. 1c, right panel), in which 4 clusters (clusters 1, 2, 3 and 4) corresponded to Mo-Macs (Fig. 1f, g) while other 2 clusters corresponded to $H2\text{-}Ab1^{hi}Ciita^{hi}H2\text{-}DMb1^{hi}Ly6c2^{int}Ccr2^{hi}Csf1r^{hi}$ monocyte-derived DCs (cluster 5) and $H2\text{-}Ab1^{hi}Ciita^{hi}H2\text{-}DMb1^{hi}Flt3^{hi}Cd209a^{hi}Tmem123^{hi}$ migratory DCs (cluster 6)[21] (Supplementary Fig. 1d). The profiles of gene expression assigned the clusters 1 and 4 to $Adgre1$(F4/80)[lo]$Ccr2$(CCR2)[hi]$Ly6c2$(Ly6C)[hi] classical monocytes and $Adgre1$(F4/80)[hi]$Ccr2$(CCR2)[lo] resident-like macrophages, respectively, while the clusters 2 and 3 were assigned to $Adgre1$(F4/80)[hi]$Ccr2$(CCR2)[hi] monocyte-derived macrophages (Fig. 1f, g). The frequency of classical monocytes (cluster 1) and monocyte-derived macrophages (clusters 2 and 3) was predominantly diminished in $Ccr2^{-/-}$ mice when compared to WT mice (Fig. 1h) in accordance with the flow cytometric analysis (Fig. 1c). Among the monocyte-derived macrophage population, the frequency of cluster 2 decreased at the transition from day 3 to day 5 post-challenge while that of cluster 3 rather increased (Fig. 1i).

### Ly6C[hi]PD-L2[lo] classical monocytes sequentially differentiate into Ly6C[lo]PD-L2[hi] macrophage via intermediate Ly6C[hi]PD-L2[hi] macrophages but not Ly6C[lo] non-classical monocytes

RNA velocity and pseudotime trajectory analyses revealed that the gene expression continuously changed in the order of cluster 1, cluster 2, cluster 3, cluster 4 (Fig. 2a, b). Considering that cluster 1 corresponds to $Ly6c2^{hi}$ classical monocytes and that cluster 2 was detected earlier than cluster 3 in the skin lesion (Fig. 1i), the differentiation of monocytes toward macrophages likely occurred in this order. Accordingly, we designated cluster 2 and 3 macrophages as early and late classical monocyte-derived macrophages (CMDMs), respectively. Along the differentiation of Mo-Mac lineage cells in the inflamed skin, the expression of $Ly6c2$ (encoding Ly6C) continuously decreased whereas that of $Adgre1$ (encoding F4/80) was gradually upregulated (Fig. 2c). The expression of $Pdcd1lg2$ (encoding PD-L2) was rarely detected in classical monocytes, upregulated in early CMDMs and then gradually downregulated thereafter (Fig. 2c).

We next sought to identify and visualize each subpopulation corresponding to clusters 1, 2, 3, and 4 by means of flow cytometry based on the surface expression of Ly6C, PD-L2 and F4/80. CD45[+]CD11b[+]SiglecF[-]Ly6G[-] Mo-Mac lineage cells present in the IgE-CAI skin lesion of WT mice were subdivided into four fractions by differential surface expression of Ly6C and PD-L2, namely Ly6C[hi]PD-L2[lo], Ly6C[hi]PD-L2[hi], Ly6C[lo] PD-L2[hi] and Ly6C[lo]PD-L2[lo] (Fig. 2d, Supplementary Fig. 2). Considering the expression profiles obtained from the scRNA-seq analysis, these four populations appeared to correspond approximately to classical monocytes (cluster 1), early CMDMs (cluster 2), late CMDMs (cluster 3) and resident-like macrophages (cluster 4), respectively. In accordance with this categorization, the majority of the latter 3 populations expressed macrophage markers[22], CD64 (FcγRI) and F4/80, on their cell surface (Supplementary Fig. 3a, b). To further validate the correspondence between each subpopulation of Mo-Macs defined based on the cell surface marker expression and each cluster defined based on the gene expression profile, we conducted bulk RNA-seq analysis of four fractions of Mo-Mac subpopulations (Supplementary Fig. 4a). Scoring of the scRNA-seq data by differentially expressed genes in each Mo-Mac population revealed that cells in clusters 1, 2, 3 and 4 indeed displayed the highest scores for Ly6C[hi]PD-L2[lo], Ly6C[hi]PD-L2[hi], Ly6C[lo] PD-L2[hi] and Ly6C[lo]PD-L2[lo] Mo-Macs, respectively (Supplementary Fig. 4b). Thus, Ly6C[hi]PD-L2[lo], Ly6C[hi]PD-L2[hi], Ly6C[lo] PD-L2[hi], and Ly6C[lo]PD-L2[lo] Mo-Macs identified by the cell surface

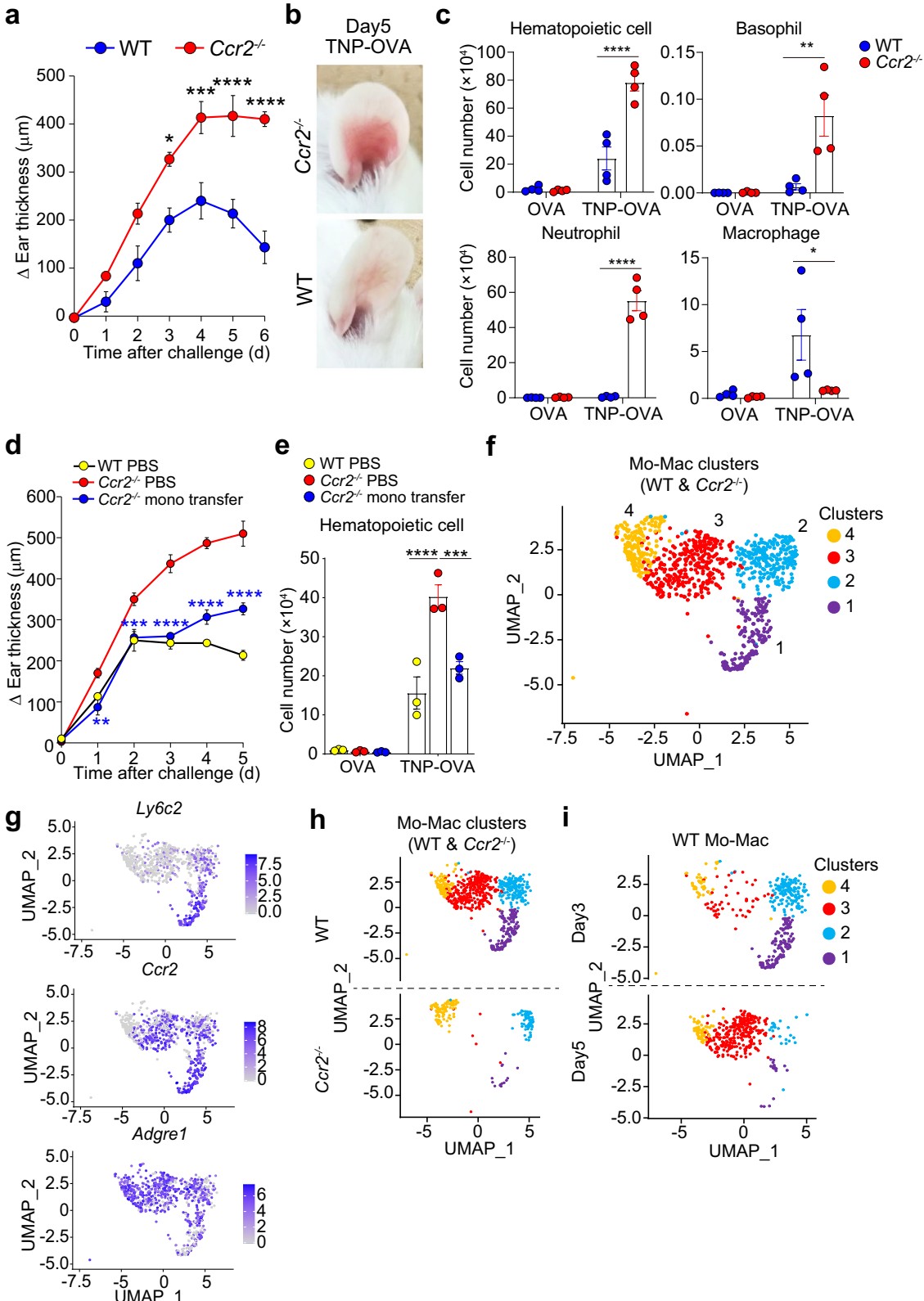

expression profile most likely correspond to classical monocytes, early CMDMs, late CMDMs and resident-like macrophages defined on the basis of the scRNA-seq data, respectively.

On day 0, Ly6C$^{lo}$PD-L2$^{lo}$ resident-like macrophages predominated among the Mo-Mac population (Fig. 2d). The frequency and the number of Ly6C$^{hi}$PD-L2$^{lo}$ classical monocytes and Ly6C$^{hi}$PD-L2$^{hi}$ early CMDMs increased until the peak of the IgE-CAI response (day 3) and

decreased thereafter (Fig. 2d, e). The number of Ly6C$^{lo}$PD-L2$^{hi}$ late CMDMs increased up to day 3 and was retained till day 5 (Fig. 2e). These observations suggested the differentiation from Ly6C$^{hi}$PD-L2$^{lo}$ classical monocytes to Ly6C$^{hi}$PD-L2$^{hi}$ early CMDMs macrophages and further into Ly6C$^{lo}$PD-L2$^{hi}$ late CMDMs. In line with this, the surface expressions of CD64 and F4/80, macrophage markers, were upregulated along this transition (Supplementary Fig. 3a, b). In order to

**Fig. 1 | Two clusters of monocyte-derived macrophages are identified in the skin lesion of IgE-CAI.** Wild-type (WT) and $Ccr2^{-/-}$ BALB/c mice were sensitized with anti-TNP IgE and challenged with intradermal administration of TNP-OVA or control OVA in the ear skin to induce IgE-CAI. **a** Time course of ear swelling (Δ ear thickness) in WT and $Ccr2^{-/-}$ mice is shown (mean ± SEM, $n = 3$ biologically independent animals for each group). $*p = 0.0145$ (for day 3), $***p = 0.0005$ (for day 4), $****p = 5.78 \times 10^{-5}$ (for day 5) and $****p = 6.35 \times 10^{-7}$ (for day 6) measured by two-way ANOVA with Sidak's multiple comparison test. **b** The gross appearance of TNP-OVA-injected ear skin on day 5 is shown. **c** The number of indicated cell types in the ear skin on day 5 is shown (mean ± SEM, $n = 4$ biologically independent animals for each group). $*p = 0.0397$ (for Mo-Mac) $**p = 0.002$ (for basophil) $****p = 4.04 \times 10^{-5}$ (for hematopoietic cell) and $p = 7.04 \times 10^{-8}$ (for neutrophil) measured by two-way ANOVA with Tukey's multiple comparison test. **d, e** CD115$^+$ monocytes isolated from the bone marrow of WT BALB/c mice or control PBS were intravenously administered to mice five times on days 0, 1, 2, 3, and 4 post-challenge. Time course of ear swelling is shown in (**d**) (mean ± SEM, $n = 3$ biologically independent animals for each group). $**p = 0.0014$ (for day 1) $***p = 0.0004$ (for day 2) $****p = 3.59 \times 10^{-9}$ (for day3), $****p = 2.30 \times 10^{-9}$ (for day 4), $****p = 1.48 \times 10^{-9}$ (for day 5) measured by two-way ANOVA with Tukey's multiple comparison test. The number of hematopoietic cells in the ear skin on day 5 is shown in (**e**) (mean ± SEM, $n = 3$ biologically independent animals for each group). $****p = 4.38 \times 10^{-5}$ $***p = 0.0008$ measured by two-way ANOVA with Tukey's multiple comparison test. **f–i** IgE-CAI was elicited in WT and $Ccr2^{-/-}$ BALB/c mice and cells isolated from the skin lesion on days 3 and 5 post-challenge was subjected to scRNA-seq analysis. UMAP plot of monocyte-macrophage (Mo-Mac) clusters (clusters 1–4) is shown in (**f**). Feature plots showing indicated gene expression in Mo-Mac clusters are shown in **g**. UMAP plots of Mo-Mac clusters in WT and $Ccr2^{-/-}$ mice are separately shown in (**h**). UMAP plots of Mo-Mac clusters in WT mice on days 3 and 5 post-challenge are separately shown in (**i**). Data shown in (**a–e**) are representative of three independent experiments. Source data are provided as a Source Data file.

experimentally prove the predicted order of the Mo-Mac differentiation, CD45.1$^+$Ly6C$^{hi}$PD-L2$^{lo}$ classical monocytes isolated from WT mice were adoptively transferred into CD45.2$^+$$Ccr2^{-/-}$ mice on day 1 post-challenge of antigens (Fig. 2f). The majority of them changed the surface phenotype from Ly6C$^{hi}$PD-L2$^{lo}$ to Ly6C$^{hi}$PD-L2$^{hi}$ (corresponding to early CMDM) on day 3, and subsequently to Ly6C$^{lo}$PD-L2$^{hi}$ (corresponding to late CMDM) and Ly6C$^{lo}$PD-L2$^{lo}$ (corresponding to resident-like macrophages) on day 5. The surface expression of F4/80 was upregulated along this transition (Supplementary Fig. 3c). On the other hand, when Ly6C$^{lo}$ non-classical monocytes were adoptively transferred into CD45.2$^+$ mice, the majority of them remained Ly6C$^{lo}$PD-L2$^{lo}$CD64$^{-/lo}$F4/80$^{-/lo}$ even on day 5 post-challenge (Supplementary Fig. 5). Thus, the combination of scRNA-seq and flow cytometric analyses enabled us to illustrate that Ly6C$^{hi}$ classical monocytes differentiate into Ly6C$^{lo}$PD-L2$^{hi}$ macrophages via intermediate Ly6C$^{hi}$PD-L2$^{hi}$ macrophages but not Ly6C$^{lo}$ non-classical monocytes.

## IL-4 receptor-mediated signaling plays an important role in the generation of early and late CMDMs from classical monocytes

We previously reported that basophil-derived IL-4 promotes the differentiation from Ly6C$^{hi}$ classical monocytes into anti-inflammatory macrophages in the IgE-CAI model[20]. This prompted us to examine whether basophil-derived IL-4 indeed drives the generation of CMDMs from classical monocytes. To this end, Ly6C$^{hi}$PD-L2$^{lo}$ classical monocytes isolated from WT mice were incubated ex vivo with culture supernatants of bone marrow-derived basophils (BMBAs) that had been stimulated with IgE plus corresponding allergens (TNP-OVA) (Fig. 2g). Nearly half of them changed their surface phenotype to Ly6C$^{hi}$PD-L2$^{hi}$ (corresponding to early CMDM) after 24h-incubation, and more than 80% of them became Ly6C$^{lo}$PD-L2$^{hi}$ (corresponding to late CMDM) after 48h-incubation. These phenotypic changes were not detected when they were incubated with culture supernatants of basophils reacted with non-relevant antigens (OVA) (Fig. 2g). Importantly, Ly6C$^{hi}$ classical monocytes deficient in IL-4 receptor failed to change their surface phenotype even when incubated with culture supernatants of allergen TNP-OVA-stimulated basophils (Fig. 2g, lower panels). This was also the case when IL-4 neutralizing antibody was added to the culture of WT monocytes (Supplementary Fig. 6a). Of note, the treatment of Ly6C$^{hi}$ classical monocytes with recombinant IL-4 promoted the surface phenotype transition into Ly6C$^{hi}$PD-L2$^{hi}$ and further Ly6C$^{lo}$PD-L2$^{hi}$, even though the ability to drive surface phenotype change appeared lower than that of culture supernatants of activated basophils (Supplementary Fig. 6b). Thus, the basophil-derived IL-4-IL-4 receptor axis appeared to contribute to the generation of CMDMs from classical monocytes. To verify these observations in vivo, we established macrophage-specific IL-4-receptor deficient mice ($Cx3cr1^{Cre/+}$ $Il4ra^{fl}$ mice). Macrophage-specific IL-4 receptor deficiency significantly impaired the transition of classical monocytes into early and late CMDMs (Supplementary Fig. 6c),

suggesting the involvement of IL-4 receptor-mediated signaling in the transition in vivo.

## Along the differentiation, CMDMs display anti-inflammatory signatures followed by metabolic rewiring concordant with their ability to efferocytosis

In accordance with the IL-4 receptor signaling-dependent differentiation, several markers of anti-inflammatory macrophages, including $Arg1$, $Mrc1$, $Cd163$, $Retnla$, $Il10$, and $Hilpda$ were upregulated while interferon-inducible genes, including $Ifit1$, $Ifit2$, $Ifit3$, $Ifitm6$, and $Mx1$, were downregulated at the transition from classical monocytes to early CMDMs (Supplementary Fig. 7a, b), suggesting the anti-inflammatory property of early CMDMs. In line with this, the gene set enrichment analysis (GSEA) revealed that genes associated with oxidative phosphorylation, a key metabolic signature for anti-inflammatory macrophages, were enriched in early CMDMs while those associated with interferon-γ response were enriched in classical monocytes (Supplementary Fig. 7c).

GSEA further revealed that genes associated with lipid catabolism and cholesterol metabolism were enriched in late CMDMs as compared to early CMDMs (Fig. 3a, Supplementary Fig. 8a, b). In line with this, late CMDMs showed higher expression of lipid-associated genes, including $Apoe$, $Abca1$, $Abcg1$ and $Trem2$, compared to classical monocytes and early CMDM (Supplementary Fig. 8c), suggesting that late CMDMs are lipid-laden cells. Indeed, Ly6C$^{lo}$PD-L2$^{hi}$ late CMDMs showed larger cell bodies with a substantial accumulation of lipid droplet-like structure in their cytoplasm, compared to Ly6C$^{hi}$PD-L2$^{lo}$ classical monocytes and resident-like macrophages (Fig. 3b). Given that TREM2$^{hi}$ lipid-associated macrophages are associated with phagocytosis[23,24], we hypothesized that late CMDMs regulate allergic inflammation through phagocytosis of dying cells. Indeed, genes associated with the recognition process of phagocytosis, including efferocytic receptors ($Mertk$, $Trem2$, and $Stab1$) and bridging molecules ($Gas6$, $Pros1$, $Mfge8$, and $C1qa$)[25], were enriched in late CMDMs as compared to early CMDMs (Fig. 3c, d, and Supplementary Fig. 8a, d). Pathway enrichment analysis also demonstrated that genes associated with lysosome and Rac1 GTPase cycles were significantly enriched in late CMDMs (Supplementary Fig. 8e), suggesting their high ability of phagocytosis. Based on these findings, we assumed that late CMDMs may dampen excess inflammation through phagocytic clearance of dying cells and perhaps allergens as well.

To address this assumption, we subcutaneously administered AcidiFluor (pH-sensitive fluorescent dye)-labeled apoptotic neutrophils or ovalbumin (OVA) as surrogate allergen to the IgE-CAI skin lesion on day 5 post-challenge and assessed the ability of efferocytosis and endocytosis in each cell type. Among cells accumulating in the IgE-CAI skin lesion, Mo-Mac lineage cells displayed predominant ability of phagocytosing both apoptotic neutrophils and antigens (Supplementary Fig. 9a, c). They showed higher efferocytic and endocytic ability on

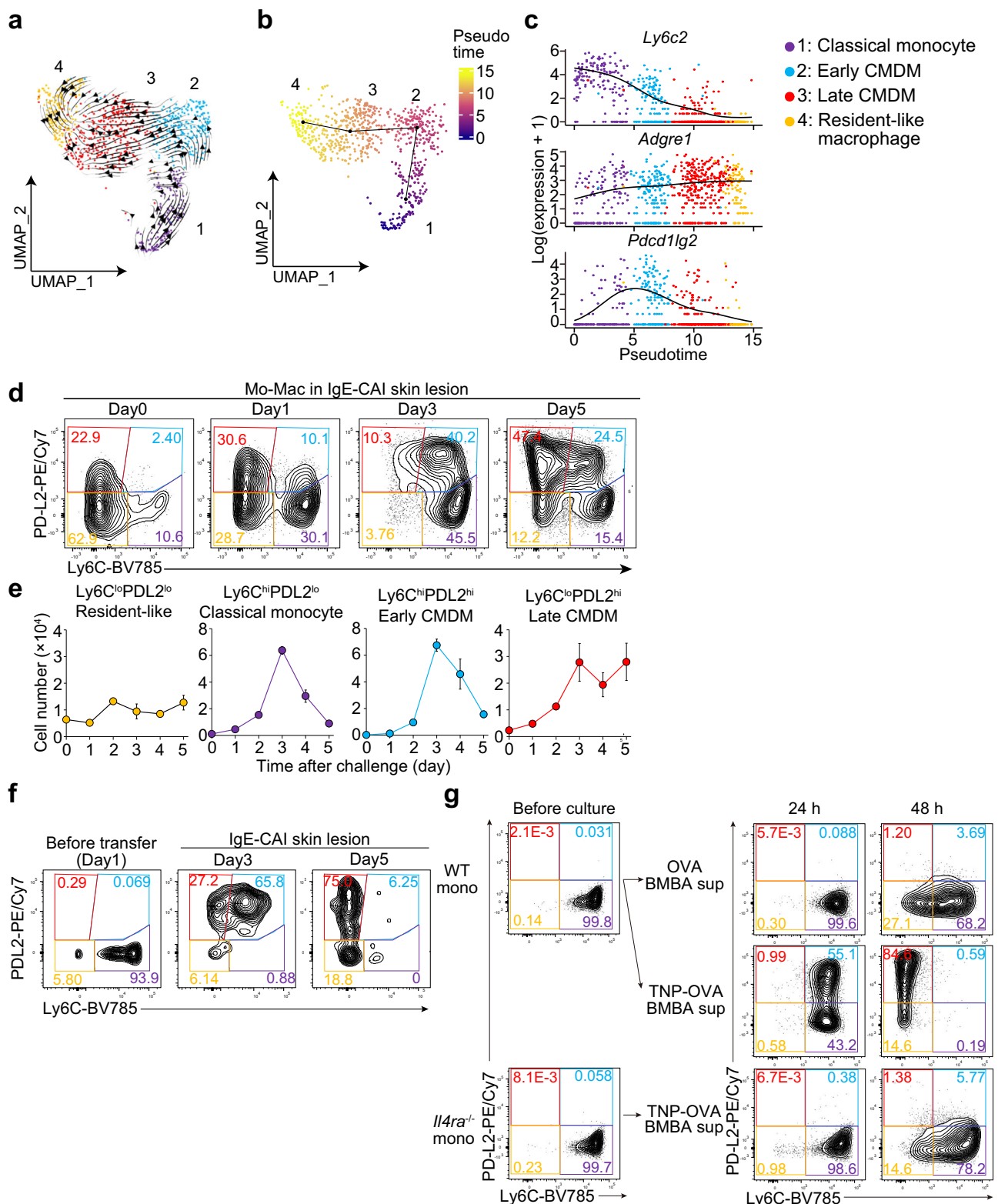

day 5 than on day 3 (Supplementary Fig. 9b, d), suggesting the high phagocytic ability of late CMDMs that expanded on day 5 (Fig. 2e). Indeed, Ly6C$^{lo}$PD-L2$^{hi}$ late CMDMs displayed enhanced phagocytic ability when compared to Ly6C$^{hi}$PD-L2$^{hi}$ early CMDMs, as judged by the frequency of AcidiFluor-positive cells and the fluorescence intensity (Fig. 3e−h).

Of note, resident-like macrophages showed reduced phagocytic ability, particularly in the phagocytosis of apoptotic neutrophils,

compared to late CMDMs (Fig. 3e−h) even though they displayed high expression of genes encoding efferocytic receptors and bridging molecules (Fig. 3d). Considering that the expression of genes associated with phagocytosis tended to decline in resident-like macrophages compared to late CMDMs (Supplementary Fig. 10a), the expression of those associated with the process following the recognition of apoptotic cells might decline in resident-like macrophages. Indeed, the expression of genes associated with PI3K-Akt

**Fig. 2 | Monocytes recruited to the IgE-CAI skin lesion sequentially differentiate into early and late CMDMs. a–c** Mo-Mac clusters shown in Fig. 1f were subjected to RNA velocity analysis. Stream plot of velocity vectors embedded in UMAP plot is shown. **b, c** Mo-Mac clusters from WT BALB/c mice shown in Fig. 1h were subjected to pseudotime trajectory analysis. UMAP plot colored by pseudotime is shown in (**b**). Gene expression changes of indicated genes along with pseudotime ordering are shown in (**c**). Colors indicate different clusters identified in Fig. 1h. **d, e** WT BALB/c mice were treated as in Fig. 1 to induce IgE-CAI. Cells isolated from the IgE-CAI skin lesion on days 0, 1, 3 and 5 were subjected to flow-cytometric analysis. In (**d**), the surface expression of PD-L2 and Ly6C in the monocyte-macrophage (Mo-Mac) populations is shown. In (**e**), time course of the number of indicated Mo-Mac subsets in the ear skin is shown (mean ± SEM, $n = 3$ biologically independent animals for Day 0–3, $n = 4$ for Day 4 and 5). **f** Ly6C[hi] monocytes isolated from the bone marrow of CD45.1[+] C57BL/6 mice were adoptively transferred to CD45.2[+]$Ccr2^{-/-}$ C57BL/6 mice on day 1 post-challenge. Cells isolated from the IgE-CAI skin lesion on days 3 and 5 were subjected to flow-cytometric analysis. The surface expression of PD-L2 and Ly6C in the CD45.1[+] Mo-Mac population is shown. **g** Ly6C[hi] classical monocytes isolated from the bone marrow of WT or $Il4ra^{-/-}$ BALB/c mice were incubated ex vivo with culture supernatants of bone marrow-derived basophils (BMBAs) that had been stimulated with anti-TNP IgE and TNP-OVA or control OVA for 18 h. The surface expression of Ly6C and PD-L2 in monocytes before the culture (left) and after 24 h- or 48 h-incubation with BMBA supernatants (middle and right, respectively) are shown. Data shown in (**d, e, g**) are representative of three independent experiments. Data shown in (**f**) are representative of two independent experiments. Source data are provided as a Source Data file.

signaling, one key signaling pathway downstream of efferocytic receptors, declined in resident-like macrophages compared to late CMDMs (Supplementary Fig. 10b).

### The accumulation of secondary necrotic neutrophils aggravates skin inflammation in $Ccr2^{-/-}$ mice

Mo-Mac populations in the IgE-CAI skin lesion of $Ccr2^{-/-}$ mice displayed impaired ability of phagocytosing apoptotic neutrophils and allergens (Fig. 3i) in accordance with the failure in the generation of CMDMs (Fig. 1h). This prompted us to investigate the functional consequence of the failure in the generation of CMDMs in vivo by using $Ccr2^{-/-}$ mice. Histopathological analysis demonstrated the presence of abscess-like leukocyte aggregates in the IgE-CAI skin lesion of $Ccr2^{-/-}$ mice but not of WT mice (Fig. 4a). Immunohistochemical analysis revealed that these leukocyte aggregates were mainly composed of Ly6G[+] neutrophils (Fig. 4b) in accordance with the exaggerated accumulation of neutrophils in the IgE-CAI skin lesion of $Ccr2^{-/-}$ mice (Fig. 1c). Adoptive transfer of WT monocytes into $Ccr2^{-/-}$ mice attenuated the formation of neutrophil-rich aggregates of leukocytes in the IgE-CAI skin lesion of $Ccr2^{-/-}$ mice, suggesting that monocyte-derived macrophages prevented the excessive neutrophil accumulation (Supplementary Fig. 11). Depletion of neutrophils by treating $Ccr2^{-/-}$ mice with either anti-Ly6G or anti-Gr-1 antibody significantly reduced the ear swelling, inflammatory cell accumulation and leukocyte aggregate formation in the skin lesion (Fig. 4c–e, and Supplementary Fig. 12), indicating that the accumulation of neutrophils contributed to the aggravated skin inflammation in $Ccr2^{-/-}$ mice.

Neutrophil-rich leukocyte clusters in the skin lesion of $Ccr2^{-/-}$ mice contained abundant dead cells with fragmented DNA (Fig. 4f) and RIPK1[+] necrotic cells (Fig. 4g), suggesting that the accumulation of TUNEL and RIPK1 double-positive secondary necrotic cells. Of note, the treatment of $Ccr2^{-/-}$ mice with RIPK1 inhibitor Nec-1s significantly suppressed ear swelling and accumulation of inflammatory cells, including neutrophils, in the IgE-CAI skin lesion (Fig. 4h, i), indicating the contribution of necrotic cells to aggravated inflammation in $Ccr2^{-/-}$ mice. These results suggested that the failure in the generation of highly phagocytic CMDMs in $Ccr2^{-/-}$ mice resulted in decreased clearance of apoptotic neutrophils while increased accumulation of necrotic neutrophils, leading to the exaggerated skin inflammation.

The treatment of $Ccr2^{-/-}$ mice with anakinra, recombinant human IL-1R antagonist, significantly reduced IgE-CAI ear swelling (Fig. 5a). Moreover, the administration of anti-IL-1α but not anti-IL-1β antibody suppressed the ear swelling as wells as the accumulation of inflammatory cells including neutrophils and ameliorated histopathological changes in $Ccr2^{-/-}$ mice (Fig. 5b–d). In line with this observation, the amounts of IL-1α protein in the IgE-CAI skin lesion were significantly higher in $Ccr2^{-/-}$ mice than in WT mice whereas the amounts of IL-1β were ~60 times lower than those of IL-1α (Supplementary Fig. 13a). Providing that the $Il1a$ mRNA expression was predominantly detected in neutrophils in $Ccr2^{-/-}$ mice (Supplementary Fig. 13b), it can be assumed that IL-1α released from dying neutrophils promotes further accumulation of neutrophils in the IgE-CAI skin lesion of $Ccr2^{-/-}$ mice, driving a vicious cycle of inflammation.

## Discussion

Recent studies reported that Ly6C[hi] classical monocytes can differentiate into not only Ly6C[hi] inflammatory but also Ly6C[lo] anti-inflammatory macrophages under certain conditions[7–10,20,26]. However, the differentiation trajectory from Ly6C[hi] classical monocytes toward Ly6C[lo] anti-inflammatory macrophages remained less clear compared to that toward Ly6C[hi] inflammatory macrophages. Some studies have proposed that Ly6C[hi] classical monocytes firstly differentiate to Ly6C[lo] non-classical monocytes and subsequently into Ly6C[lo] macrophages in murine models of schistosome-infected liver and sterile liver injury[15,16]. Other studies showed that Ly6C[lo] macrophages can be generated from Ly6C[hi] classical monocytes even in the absence of blood-circulating Ly6C[lo] non-classical monocytes in models of skeletal muscle injury and myocardial infarction[17,18]. The present study with scRNA-seq, flow cytometric and functional analyses clearly demonstrated in a skin allergy model that Ly6C[hi]PD-L2[lo] classical monocytes differentiate to Ly6C[lo]PD-L2[hi] pro-resolving macrophages (late CMDMs) in a stepwise manner, namely via intermediate Ly6C[hi]PD-L2[hi] macrophages (early CMDMs) but not Ly6C[lo] non-classical monocytes, in an IL-4 receptor-dependent manner. This appears to accord with the recently proposed model of stepwise differentiation of tissue-infiltrating monocytes[27].

Along the differentiation from classical monocytes to early CMDMs, the expression of not only $Adgre1$ encoding F4/80, typical marker of macrophages, but also genes linked to anti-inflammatory functions was upregulated, most likely through IL-4 receptor-mediated signaling. By contrast, the expression of interferon-inducible genes was downregulated, indicating the functional switch from inflammatory monocytes to anti-inflammatory macrophages. Along the transition from early to late CMDMs, the expression of genes encoding efferocytosis receptors and bridging molecules as well as those associated with phagocytosis was upregulated. Moreover, late CMDMs displayed the enhanced expression of genes associated with lipid catabolism and cholesterol metabolism, implying that late CMDMs are equipped with the machinery to catabolize lipid components derived from dying cells taken up by efferocytosis[25,28]. In accordance with such gene expression profiling, CMDMs, particularly late ones, possess high ability to phagocytose apoptotic neutrophils in vivo, and late CMDMs contain lipid droplet-like structures, resembling foam cells observed in atherosclerosis[29,30]. Taken together, the differentiation of skin-recruited classical monocytes into CMDMs, particularly late ones, appears to be a crucial step for the resolution of allergic inflammation by means of phagocytic clearance of apoptotic cells.

van Dierendonck et al. reported that the lipid droplet accumulation in macrophages suppresses the production of inflammatory mediators, including IL-6 and PGE2, in a manner dependent on the upregulation of the hypoxia-inducible lipid droplet-associated (HILPDA) protein[31]. In our study, the upregulation of $Hilpda$ was observed at the stage of early CMDMs (Supplementary Fig. 7a),

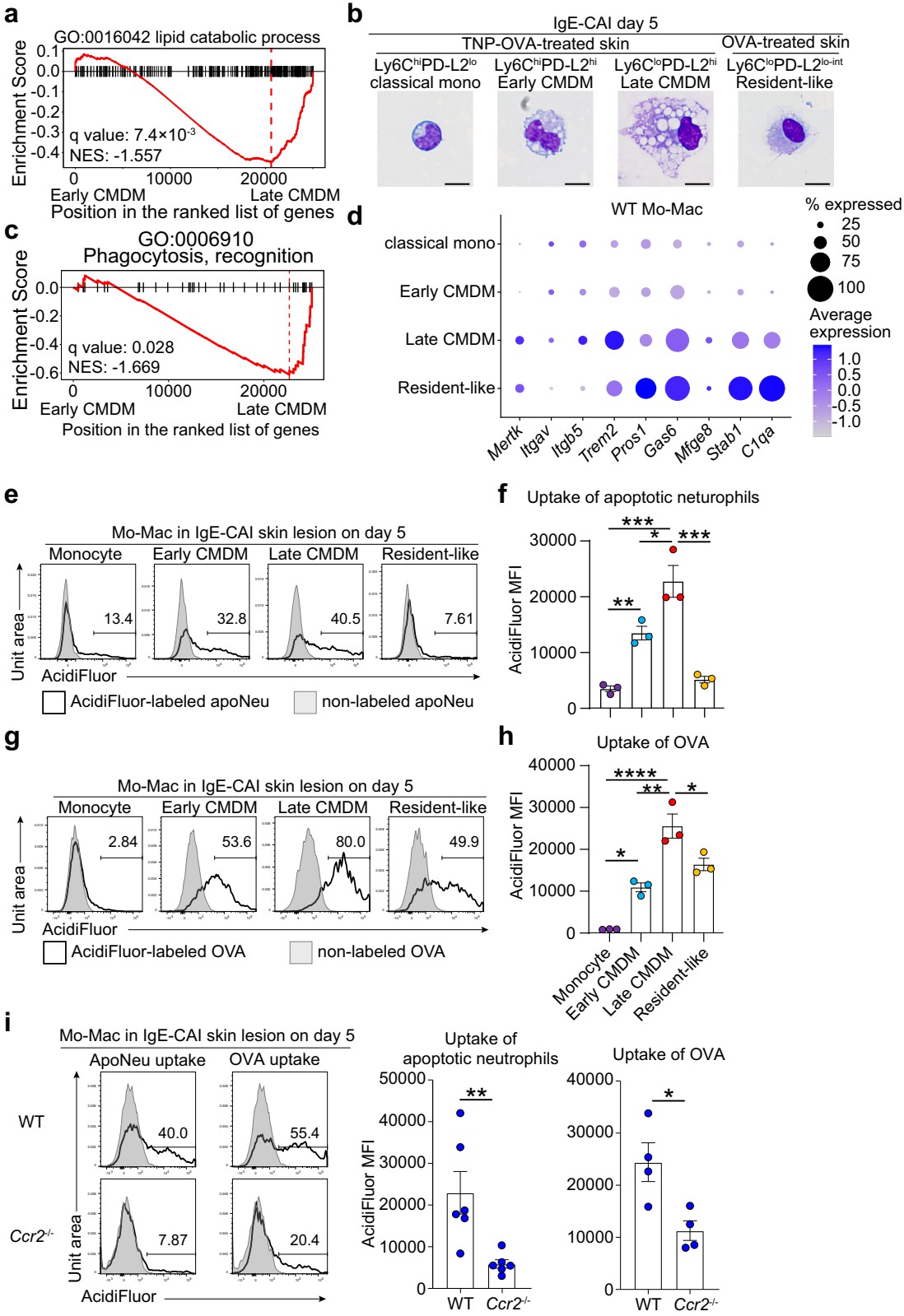

suggesting the possibility that the lipid droplet accumulation may start even prior to the stage of late CMDMs, contributing to the resolution of inflammation through the suppression of inflammatory mediators. We also identified that late CMDMs in the allergic skin lesion display high expression of the gene encoding TREM2. *Trem2*[hi] lipid-associated macrophages have been shown to regulate inflammatory responses in certain disorders, including metabolic disorders, neurodegenerative

diseases, and tumors[23,32–34]. *Trem2*[hi] macrophages expressing both lipid-associated genes and efferocytosis-associated genes are also detected in nerve injury model[12]. In a model of myocardial infarction, *Ly6c2*[hi] classical monocytes differentiate sequentially to early macrophages, transient macrophages and *Trem2*[hi] late macrophages, leading to cardiac tissue remodeling[35]. Thus, the stepwise differentiation trajectory from classical monocyte into TREM2[hi] lipid-associated

**Fig. 3 | Late CMDMs display higher efferocytic ability than do early CMDMs.**
Early and late CMDMs were compared by using GSEA. Enrichment plot of genes involved in lipid catabolic process (**a**) and phagocytosis (**c**). **b** WT BALB/c mice were treated as in Fig. 1 to induce IgE-CAI. Classical monocytes, early CMDMs and late CMDMs isolated from TNP-OVA-treated ear skins, and resident macrophages isolated from OVA-treated ear skins were subjected to Diff-Quik staining. Bars indicate 10 μm. **d** Dot plot showing the expression level of indicated genes in classical monocytes, early CMDMs, late CMDMs and resident-like macrophages. **e–h** WT BALB/c mice were treated as in Fig. 1 to induce IgE-CAI. Apoptotic neutrophils (**e**, **f**) or OVA (**g**, **h**) labeled (open histograms) or unlabeled (shaded histograms) with AcidiFluor were intradermally administered to the ear skin on day 5 post-challenge. Two hours after the injection, skin samples were subjected to flow cytometric analysis. In (**e**) and (**g**), histograms of AcidiFluor fluorescence in Mo-Mac subsets are shown. In (**f**) and (**h**), the mean fluorescence intensity (MFI) of AcidiFluor in Mo-Mac subsets is shown (mean ± SEM, $n = 3$ biologically independent animals each). In (**f**), **$**p = 0.0096$ (mono vs. early CMDM); ***$p = 0.0001$ (mono vs. late CMDM) *$p = 0.0144$ (early vs. late CMDM), ***$p = 0.0002$ (late CMDM vs. resident-like) measured by one-way ANOVA with Tukey's multiple comparison test. In (**h**), *$p = 0.0135$ (mono vs. early CMDM); ****$p = 3.36 \times 10^{-5}$ (mono vs. late CMDM) **$p = 0.0014$ (early vs. late CMDM), *$p = 0.0218$ (late CMDM vs. resident-like) measured by one-way ANOVA with Tukey's multiple comparison test. **i** WT and $Ccr2^{-/-}$ BALB/c mice were treated as in (**e**) and (**g**). Histograms of AcidiFluor fluorescence in total Mo-Macs are shown in left panels. The MFI of AcidiFluor in Mo-Mac populations is shown in right panels (mean ± SEM, $n = 6$ and $n = 4$ biologically independent animals for apoptotic neutrophil uptake and OVA uptake, respectively). **$p = 0.0082$ (for apoptotic neutrophil) *$p = 0.0196$ (for OVA) measured by two-sided unpaired Student's $t$ test. Data shown in (**b**, **e–i**) are representative of three independent experiments. Source data are provided as a Source Data file.

macrophages might be a shared pathway in the resolution of inflammation and the remodeling after tissue injury.

The present study clearly demonstrated that IL-4 derived from basophils is an important factor driving the differentiation of classical monocytes to early and late CMDMs, leading to the resolution of allergic inflammation through the clearance of apoptotic cells by phagocytic CMDMs. Considering that some of IL-4 receptor-deficient monocytes displayed the surface phenotype similar to that of CMDMs to a certain extent in the IgE-CAI skin lesion (Supplementary Fig. 6c), factors other than IL-4 may also contribute to the differentiation to CMDMs. It has been reported that the efferocytosis of apoptotic cells promotes the differentiation of monocytes to remodeling macrophages in cooperation with IL-4 and IL-13[36]. Indeed, the 24h- and 48h-incubation of Ly6C[hi]PD-L2[lo] classical monocytes ex vivo with apoptotic neutrophils induced the change in the surface phenotype to Ly6C[hi]PD-L2[hi] and Ly6C[lo]PD-L2[lo], respectively (Supplementary Fig. 14), suggesting that the engulfment of apoptotic neutrophils may also promote the differentiation into early and late CMDMs.

The examination of the IgE-CAI skin lesion of $Ccr2^{-/-}$ mice illustrated the functional consequence of the failure in the generation of CMDMs, namely the appearance of abscess-like clusters of TUNEL and RIPK1 double-positive neutrophils. Depletion of neutrophils or treatment with RIPK1 inhibitor ameliorated excess inflammation in $Ccr2^{-/-}$ mice, suggesting that the accumulation of secondary necrotic neutrophils exaggerated skin inflammation. We identified IL-1α but not IL-1β, most likely released from dying neutrophils, is a key driver of the vicious cycle of inflammation in $Ccr2^{-/-}$ mice, leading to the formation of abscess-like neutrophil clusters in the skin lesion, in contrast to the case of the abscess formation during *Staphylococcus aureus* infection where IL-1β plays a critical role[37]. The scRNA-seq analysis suggested that the genes encoding IL-1α receptor and neutrophil-attracting chemokine CXCL5 are predominantly expressed by skin *Pdgfra*+ fibroblasts (Supplementary Fig. 15a, b). The treatment of $Ccr2^{-/-}$ mice with IL-1α neutralizing antibody attenuated the expression of *Cxcl5* in skin fibroblasts (Supplementary Fig. 15c), suggesting that skin fibroblasts could be the target of IL-1α and produces CXCL5 to promote neutrophil infiltration into the IgE-CAI skin lesion. Taken together, in $Ccr2^{-/-}$ mice, the failure in the generation of highly efferocytic CMDMs appeared to cause the accumulation of secondary necrotic neutrophils that in turn produce IL-1α, promoting further recruitment of neutrophils through the production of neutrophil-attracting chemokines by IL-1α-stimulated skin fibroblasts. In WT mice, highly efferocytic CMDMs efficiently clears apoptotic cells, thus preventing the IL-1α-mediated aggravation of neutrophilic inflammation. Of note, the exaggerated neutrophil accumulation in $Ccr2^{-/-}$ mice has been reported in other inflammatory settings, including infection and injury models[38–43], suggesting that the IL-1α-mediated vicious cycle of inflammation identified in the present study may be operative in these settings as well.

In conclusion, we took advantage of single-cell transcriptomics in combination with flow cytometric and functional analyses in the present study and clarified the stepwise differentiation from Ly6C[hi] classical (inflammatory) monocytes toward Ly6C[lo] pro-resolving CMDMs in the IgE-CAI skin allergy model. The failure of this process in the IgE-CAI response drives the IL-1α-mediated vicious cycle of inflammation, leading to neutrophilic aggravation of the skin inflammation. Further studies are needed to clarify whether the similar process is operative in the resolution phase of other inflammatory disorders.

## Methods

### Mice
BALB/c (BALB/cCrSlc) and C57BL/6J (C57BL/6JmsSlc) mice were purchased from Sankyo Labo Service Corporation, Inc, Japan. CD45.1 congenic C57BL/6 (B6.SJL-*Ptprc*$^a$*Pepc*$^b$/BoyJ) mice, *Cx3cr1*[Cre] C57BL/6 (B6J.B6N(Cg)-*Cx3cr1*[tm1.1(cre)Jung]/J) mice[44] and *Il4ra*$^{-/-}$ BALB/c (BALB/c-*Il4ra*[tm1Sz]) mice[45] were purchased from the Jackson Laboratory. *Il4ra*[fl] (*Il4ra*[tm2Fbb]) C57BL/6 mice[46] were previously established. *Cx3cr1*[Cre] C57BL/6 and *Il4ra*[fl] C57BL/6 mice were cross-bred in our laboratory to establish *Cx3cr1*[Cre] *Il4ra*[fl] C57BL/6 mice that are deficient for IL-4 receptor only in Mo-Mac lineage cells. *Ccr2*$^{-/-}$ BALB/c (*Ccr2*[tm1Mae]) mice[47] were kindly provided by W.A. Kuziel (External Scientific Affairs, Daiichi Sankyo Group, Edison, NJ) and N. Mukaida (Kanazawa University). *Ccr2*$^{-/-}$ C57BL/6 mice were generated by N. Mukaida (Kanazawa University) by backcrossing *Ccr2*$^{-/-}$ BALB/c mice to C57BL/6 strain. Mice were maintained under specific pathogen-free conditions in our animal facilities. Animal rooms are maintained at $22 \pm 3$ °C, with a 12 h:12 h light-dark cycle, and humidity is maintained between 30% and 70%. Animals are housed in individually ventilated cages (Innovive Caging System) and are fed with natural ingredient chow diet ad libitum (Japan CLEA: Rodent Diet CE-2). Cages are bedded with cloth material (Japan SLC: Q-pura chip). Seven-to-twelve-week-old male mice were used in the study. Animals are group-housed whenever possible. At the end of all experiments, mice were euthanized by $CO_2$ inhalation. All animal studies were approved by the Institutional Animal Care and Use Committee of Tokyo Medical and Dental University (No. A2022-023C2).

### Antibodies
The following antibodies were purchased from BioLegend: Alexa Fluor 488-conjugated anti-CD45.2 (clone: 104, catalog#:109816, dilution 1:400), APC-conjugated anti-CD200R3 (clone: Ba13, catalog#: 142208, dilution 1:400), anti-CD64 (clone: X54-5/7.1, catalog#:139306, dilution 1:400), APC-Cy7-conjugated anti-Ly6G (clone: 1A8, catalog#: 127624, dilution 1:400), APC/Fire 750-conjugated anti-Ly6G (clone: 1A8, catalog#: 127652, dilution 1:400), FITC-conjugated anti-CD45 (clone: 30-F11, catalog#: 103108, dilution 1:400), PacificBlue-conjugated anti-c-Kit (clone: 2B8, catalog#: 105820, dilution 1:400), anti-CD45.1 (clone: A20, catalog#: 110722, dilution 1:400), BV421-conjugated PD-L2 (clone: TY25, catalog#: 107219, dilution 1:400), anti-CD11b (clone: M1/70,

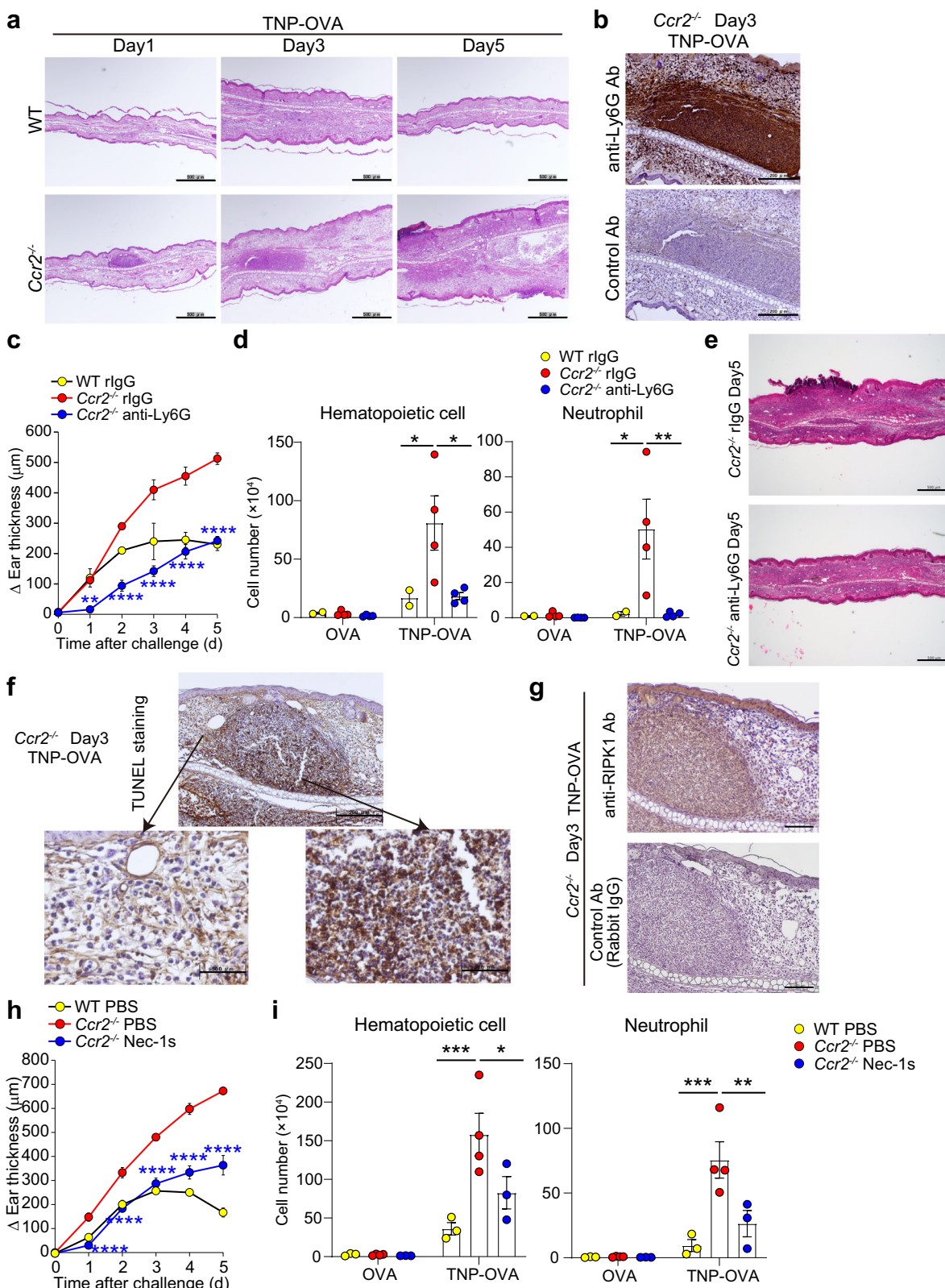

catalog#: 101251, dilution 1:400), BV510-conjugated anti-CD45.1 (clone: A20, catalog#: 110741, dilution 1:400), BV605-conjugated anti-CD11b (clone: M1/70, catalog#: 101257, dilution 1:400), BV711-conjugated anti-F4/80 (clone: BM8, catalog#: 123147, dilution 1:400), BV785-conjugated anti-Ly6C (clone: HK1.4, catalog#: 128041, dilution 1:400), PE-conjugated anti-PD-L2 (clone: TY25, catalog#: 107206, dilution 1:400), PE-Cy7-conjugated anti-CD49b (clone: HMα2, catalog#: 103518,

dilution 1:400), anti-F4/80 (clone: BM8, catalog#:123114, dilution 1:400), anti-PD-L2 (clone: TY25, catalog#: 107214, dilution 1:400). The following antibodies were purchased from BD Biosciences: Alexa 647-conjugaed anti-Siglec-F (clone: E50-2440, catalog#: 562680, dilution 1:400), BV421-conjugated anti-Siglec-F (clone: E50-2440, catalog#: 562681, dilution 1:400), BV480-conjugated c-Kit (clone: 2B8, catalog#: 566074, dilution 1:400). IgE mAb specific to 2,4,6-trinitrophenol (TNP)

**Fig. 4 | Accumulation of necrotic neutrophils aggravates skin inflammation in *Ccr2*$^{-/-}$ mice.** WT and *Ccr2*$^{-/-}$ BALB/c mice were treated as in Fig. 1 to induce IgE-CAI. **a** Ear samples prepared on days 1, 3, and 5 were subjected to HE staining. Bars indicate 500 μm. **b** Ear specimens collected from *Ccr2*$^{-/-}$ BALB/c mice on day 3 post-challenge were subjected to immunostaining with anti-Ly6G antibody or its isotype-matched control antibody. Data are representative of three independent experiments. Bars indicate 200 μm. **c–e** Neutrophil depletion antibody (anti-Ly6G) or control antibody (rIgG) was intraperitoneally administered to mice on days 0, 1, 2, 3, and 4 post-challenge. In (**c**), time course of ear swelling (Δ ear thickness) is shown (mean ± SEM, *n* = 2, *n* = 4 and *n* = 4 biologically independent animals for WT rIgG group, *Ccr2*$^{-/-}$ rIgG group and *Ccr2*$^{-/-}$ anti-Ly6G group, respectively). **\*\****p* = 0.0037 (for day 1), **\*\*\*\****p* = 2.93 × 10$^{-7}$ (for day 2), **\*\*\*\****p* = 2.09 × 10$^{-10}$ (for day 3), **\*\*\*\****p* = 1.37 × 10$^{-8}$ (for day 4), and **\*\*\*\****p* = 3.02 × 10$^{-9}$ (for day 5) measured by two-way ANOVA with Tukey's multiple comparison test. In d, the number of hematopoietic cells and neutrophils in the ear skin on day 5 is shown (mean ± SEM; *n* = 2, *n* = 4, and *n* = 4 biologically independent animals for WT rIgG group, *Ccr2*$^{-/-}$ rIgG group and *Ccr2*$^{-/-}$ anti-Ly6G group, respectively). In the left panel, \**p* = 0.042 (WT rIgG vs. *Ccr2*$^{-/-}$ rIgG), \**p* = 0.0125 (*Ccr2*$^{-/-}$ rIgG vs. *Ccr2*$^{-/-}$ anti-Ly6G) measured by two-way ANOVA with Tukey's multiple comparison test. In the right panel \**p* = 0.0325 (WT rIgG vs. *Ccr2*$^{-/-}$ rIgG), \**p* = 0.0071 (*Ccr2*$^{-/-}$ rIgG vs. *Ccr2*$^{-/-}$ anti-Ly6G) measured by two-way ANOVA with Tukey's multiple comparison test. In (**e**), HE-stained ear

specimens collected on day 5 are shown. Bars indicate 500 μm. Ear specimens collected from *Ccr2*$^{-/-}$ mice on day 3 post-challenge were subjected to TUNEL staining (**f**), immunostaining with anti-RIPK1 antibody (**g**) or its isotype-matched control antibody. Data are representative of three independent experiments. Bars in (**f**) indicate 200 (upper panel) and 50 μm (lower panels), respectively. Bars in (**g**) indicate 100 μm. **h, i** RIPK1 inhibitor (Necrostatin-1s; Nec-1s) or control PBS was intraperitoneally administered to mice on days 0, 1, 2, 3, and 4 post-challenge. In (**h**), time course of ear swelling (Δ ear thickness) is shown (mean ± SEM, *n* = 3, *n* = 4, and *n* = 3 biologically independent animals for WT PBS group, *Ccr2*$^{-/-}$ PBS group and *Ccr2*$^{-/-}$ Nec-1s group, respectively). **\*\*\*\****p* = 6.12 × 10$^{-5}$ (for day 1), **\*\*\*\****p* = 8.96 × 10$^{-7}$ (for day 2), **\*\*\*\****p* = 2.43 × 10$^{-10}$ (for day 3), **\*\*\*\****p* = 1.42 × 10$^{-12}$ (for day 4), and **\*\*\*\****p* = 1.06 × 10$^{-12}$ (for day 5) measured by two-way ANOVA with Tukey's multiple comparison test. In (**i**), the number of hematopoietic cells and neutrophils in the ear skin on day 5 is shown (mean ± SEM, *n* = 3, *n* = 4, and *n* = 3 biologically independent animals for WT PBS group, *Ccr2*$^{-/-}$ PBS group and *Ccr2*$^{-/-}$ Nec-1s group, respectively) In the left panel, \*\*\**p* = 0.001 (WT PBS vs. *Ccr2*$^{-/-}$ PBS), \**p* = 0.043 (*Ccr2*$^{-/-}$ PBS vs. *Ccr2*$^{-/-}$ Nec-1s) measured by two-way ANOVA with Tukey's multiple comparison test. In the right panel \*\*\**p* = 0.0005 (WT PBS vs. *Ccr2*$^{-/-}$ PBS), \*\**p* = 0.0075 (*Ccr2*$^{-/-}$ PBS vs. *Ccr2*$^{-/-}$ Nec-1s) measured by two-way ANOVA with Tukey's multiple comparison test. Data shown in (**a–i**) are representative of three independent experiments. Source data are provided as a Source Data file.

---

was prepared from hybridoma (IGELb4, catalog#: TIB141; ATCC) in our laboratory.

## Induction of IgE-CAI

To elicit IgE-CAI[20,48], mice were first sensitized with intravenous injection of 300 μg of anti-TNP IgE, and on the following day challenged with an intradermal injection of 10 μg TNP$_{12}$-conjugated ovalbumin (OVA) and control OVA into the right and left ear, respectively. The value of Δear thickness, the differences in ear thickness (right−left) was calculated for the evaluation of inflammation.

## Administration of antibodies and inhibitors to mice

In neutrophil depletion experiment, mice were intraperitoneally administered with 500 μg/day of Ultra-LEAF anti-mouse Ly6G antibody (clone: 1A8; catalog#: 127650; BioLegend), 250 μg/day anti-mouse Gr-1 antibody (clone: RB6-8C5; prepared in our laboratory from hybridoma), or control rat IgG (Jackson ImmunoReseach) once a day during the development of IgE-CAI. For inhibition of necrosis, mice were intravenously administered with 150 μg/day of Nec-1s (7-Cl-O-Nec1; Calbiochem), or vehicle alone (PBS containing DMSO) once a day during the development of IgE-CAI. For IL-1 neutralization experiment, mice were intraperitoneally administered with 1 mg/day of Anakinra (Raleukin; MedChemExpress), 300 μg/day of anti-IL-1α antibody (clone: ALF-161; BioXCell catalog#: BE0243), 300 μg/day of anti-IL-1β antibody (clone: B122; BioXCell catalog#: BE0246), or 300 μg/day polyclonal Armenian hamster IgG antibody (BioXCell catalog#: BE0091) once a day during the development of IgE-CAI.

## Monocyte transfer experiment

Monocytes were enriched from bone marrow cells by using EasySep Mouse Monocyte Isolation Kit (StemCell Technologies, catalog#: 19861) according to manufacturer's protocol. Bone marrow monocytes (1 × 10$^6$ cells/injection/mouse) or control PBS were intravenously administered to the mice once a day during the development of IgE-CAI.

For fate-tracking experiments and ex vivo monocyte culture experiments, monocytes were first enriched from the bone marrow of CD45.1 congenic C57BL/6 mice or BALB/c mice by using EasySep Mouse Monocyte Isolation Kit (StemCell Technologies, catalog#: 19861). Ly6C$^{hi}$ monocytes were isolated from monocytes by using biotinylated anti-Ly6C antibody (clone: AL-21, catalog#: 557359, dilution 1:400; BD Biosciences) and EasySep Mouse biotin positive selection Kit II (StemCell Technologies, catalog#: 17665). Ly6C$^{lo}$ monocytes

were isolated from monocytes by depleting Ly6C$^+$ population using biotinylated anti-Ly6C antibody (clone: AL-21, BD Biosciences) and EasySep Mouse Streptavidin RapidSpheres Isolation Kit (StemCell Technologies, catalog#: 19860).

## Ex vivo culture of monocytes

Cells were cultured in RPMI complete medium (RPMI 1640 medium (Nacalai tesque) supplemented with 10% fetal bovine serum (Sigma-Aldrich), 100 U/mL penicillin (Nacalai tesque), 100 μg/mL streptomycin (Nacalai tesque), 1 mM sodium pyruvate (Nacalai tesque), 0.1 mM nonessential amino acids (Nacalai tesque), and 5 × 10$^{-5}$ M 2-mercaptoethanol (Gibco)). Bone marrow-derived basophils (BMBAs) were generated by culturing bone marrow cells in the presence of murine IL-3 (300 pg/mL; BioLegend) for 6 days[49]. BMBAs were then sensitized with anti-TNP IgE antibody (clone: IgE-Lb4: 1 μg/mL) for 24 h, followed by the magnetic enrichment of CD49b$^+$ fractions by using biotinylated CD49b antibody (clone: DX5, catalog#: 108904, dilution 1:400; BioLegend) and MojoSort Streptavidin Nanobeads (BioLegend). BMBAs were then incubated with TNP-OVA or control OVA (10 ng/mL each) for 18 h. Ly6C$^{hi}$ monocytes isolated from the bone marrow of BALB/c mice or *Il4ra*$^{-/-}$ mice were incubated for 24 h or 48 h with the culture supernatant of BMBAs. In some experiments, Ly6C$^{hi}$ monocytes (1 × 10$^5$ cells/well) from WT mice were incubated for 24 h or 48 h with recombinant mouse IL-4 (20 ng/mL; BioLegend), apoptotic neutrophils (1 × 10$^6$ cells/well), or control PBS.

## Histopathological analysis

For hematoxylin and eosin staining, ear specimens were fixed with 4% paraformaldehyde/PBS solution, embedded in paraffin, cut into 5 μm-thick sections, and stained with hematoxylin and eosin (Muto pure chemicals). For immunohistochemical staining, ear sections were treated with microwaves, followed by incubation with methanol containing 0.3% H$_2$O$_2$, avidin/biotin blocking kit (Vector laboratories, catalog#: SP-2001), and normal serum (Vector laboratories). Ear sections were then incubated with the following antibodies at 4 °C for 18 h: rat anti-Ly6G antibody (1 μg/mL; clone: 1A8, catalog#: 127602; BioLegend), rat IgG2a isotype control antibody (1 μg/mL; clone: RTK2758, catalog#: 400502; BioLegend), rabbit polyclonal anti-RIPK1 antibody (2.5 μg/mL, catalog#: NBP1-77077, Novus Biologicals), and rabbit polyclonal IgG isotype control antibody (2.5 μg/mL, catalog#: ab37415, Abcam). They were incubated subsequently with biotinylated secondary antibody, ABC reagent (Vector laboratories), 3′-diaminobenzidene tetrahydrochloride solution (Nacalai tesque),

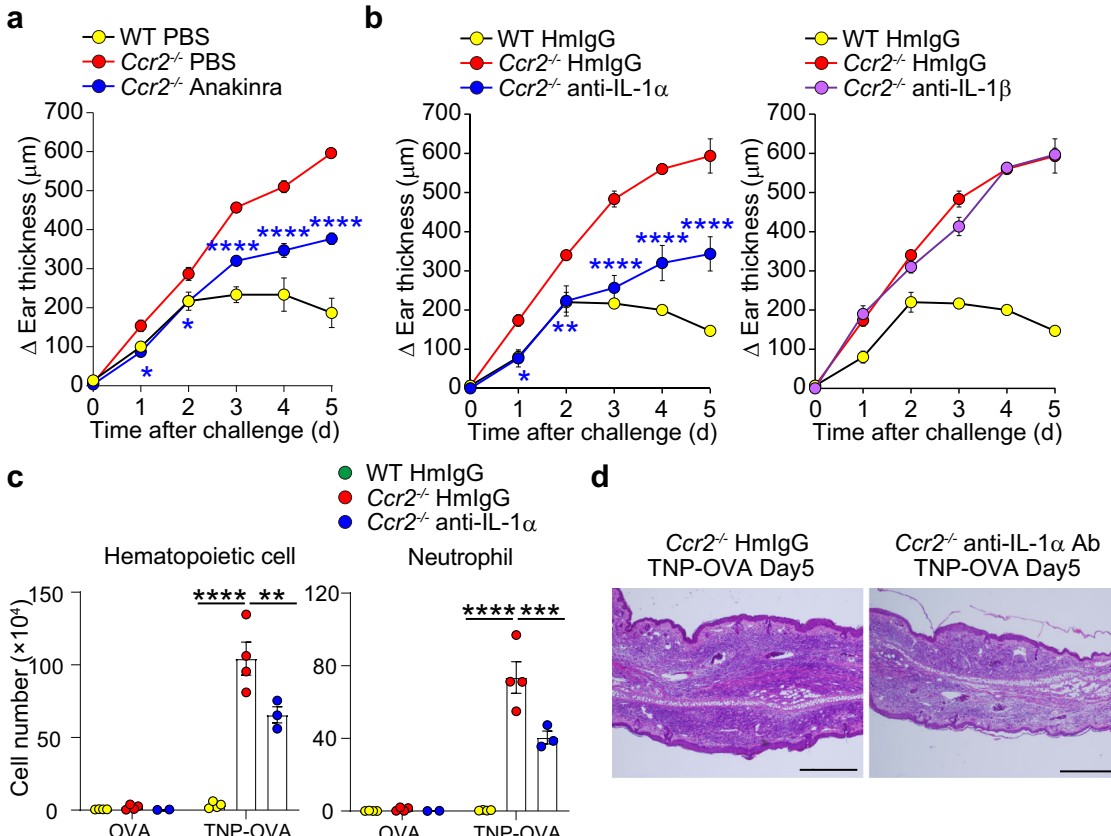

**Fig. 5 | IL-1α contributes to the neutrophil accumulation in the IgE-CAI skin lesion of *Ccr2*⁻/⁻ mice.** WT and *Ccr2*⁻/⁻ BALB/c mice were treated as in Fig. 1 to induce IgE-CAI. **a** Anakinra (human recombinant IL-1 antagonist) or control PBS was intraperitoneally administered to indicated mice on days 0, 1, 2, 3, and 4 post-challenge. The time course of ear swelling (Δ ear thickness) is shown (mean ± SEM, *n* = 3 biologically independent animals for each group). *$p$ = 0.0371 (for day 1), *$p$ = 0.0274 (for day 2), ****$p$ = 1.89×10⁻⁵ (for day 3), ****$p$ = 7.98 × 10⁻⁷ (for day 4), and ****$p$ = 1.19 × 10⁻⁹ (for day 5) measured by two-way ANOVA with Tukey's multiple comparison test. **b** Neutralizing antibody against IL-1α, IL-1β or its isotype-matched control antibody (HmIgG) was intraperitoneally administered to indicated mice on days 0, 1, 2, 3, and 4 post-challenge. Time course of ear swelling (Δ ear thickness) is shown (mean ± SEM, *n* = 3 biologically independent animals for each group). In the left panel, *$p$ = 0.0219 (for day 1), **$p$ = 0.0049 (for day 2), ****$p$ = 3.74 × 10⁻⁷ (for day 3), ****$p$ = 1.16 × 10⁻⁷ (for day 4), and ****$p$ = 4.86 × 10⁻⁸ (for day 5) measured by two-

way ANOVA with Tukey's multiple comparison test. **c** The number of hematopoietic cells and neutrophils in the ear skin of indicated mice on day 5 is shown (mean ± SEM, *n* = 4 biologically independent animals for WT HmIgG and *Ccr2*⁻/⁻ HmIgG groups; *n* = 3 biologically independent animals for TNP-OVA-treated *Ccr2*⁻/⁻ anti-IL-1α group; *n* = 2 biologically independent animals for OVA-treated *Ccr2*⁻/⁻ anti-IL-1α group. In the left panel, ****$p$ = 1.51 × 10⁻⁸ (WT HmIgG vs. *Ccr2*⁻/⁻ HmIgG), **$p$ = 0.0032 (*Ccr2*⁻/⁻ HmIgG vs. *Ccr2*⁻/⁻ anti-IL-1α) measured by two-way ANOVA with Tukey's multiple comparison test. In the right panel ****$p$ = 2.46 × 10⁻⁸ (WT HmIgG vs. *Ccr2*⁻/⁻ HmIgG), ***$p$ = 0.001 (*Ccr2*⁻/⁻ HmIgG vs. *Ccr2*⁻/⁻ anti-IL-1α) measured by two-way ANOVA with Tukey's multiple comparison test. **d** HE stained ear specimens collected from indicated mice on day 5 are shown. Bars indicate 500 μm. Data shown in (**a–d**) are representative of three independent experiments. Source data are provided as a Source Data file.

and counterstained with hematoxylin. TUNEL staining was conducted by using TACS 2 TdT-DAB In Situ Apoptosis Detection Kit (R&D systems, catalog#: 4810-30-K), according to manufacturer's protocols.

**Flow cytometric analyses and cell sorting**

For flow cytometric analyses, single-cell suspensions were prepared from the ear skin by treating excised ears with collagenase (125 U/mL, Wako) in RPMI complete medium at 37 °C for 2 h, followed by depletion of red blood cells. After pre-incubation with TruStain FcX PLUS antibody (anti-CD16/32 antibody, clone: S17011E, catalog#: 156604, dilution 1:200; BioLegend) and normal rat serum (Merck Millipore, dilution 1:10) on ice for 10 min to prevent the non-specific binding of irrelevant Abs, cells were stained with indicated combination of Abs, and analyzed with FACSLyric (BD Biosciences) or sorted with FACSAriaIII (BD Biosciences). Each cell lineage was identified as follows: neutrophils (CD45⁺ Ly6G⁺ Siglec-F⁻), eosinophils (CD45⁺ Ly6Gⁱⁿᵗ Siglec-F⁺), basophils (CD45ⁱⁿᵗ c-Kit⁻ CD49b⁺ CD200R3⁺), Mo-Macs (CD45⁺ Ly6G⁻ Siglec-F⁻ CD11b⁺ F4/80⁺), and non-hematopoietic cells (CD45⁻). For experiments in Fig. 3b, sort-purified Mo-Mac subpopulations were

stained by using Diff-Quik stain (Sysmex, catalog#: 16920), according to the manufacturer's protocol.

**Quantitative PCR**

cDNA was synthesized from sort-purified fibroblasts by using QuantAccuracy RT-RamDA cDNA Synthesis Kit (TOYOBO), according to manufacturer's protocol. Quantitative PCR of the cDNA was performed on StepOnePlus Real-Time PCR system (Applied Biosystems) using a Fast SYBR Green Master Mix (Applied Biosystems) and the following primer sets: *Cxcl5* (sense-GCATTTCTGTTGCTGTTCACGCTG, anti-sense-CCTCCTTCTGGTTTTTTCAGTTTAGC), *RplpO* (sense- GCCCCTG CACTCTCGCTTTC, antisense- TGCCAGGACGCGCTTGT). Relative gene expression levels were calculated by ΔΔCT method with normalization using *RplpO* expression.

**Measurement of cytokines and chemokine in the skin lesions**

The ear skins suspended in PBS containing 0.1% Triton X-100 (Sigma-Aldrich) and Halt protease and phosphatase inhibitor cocktail (Thermo Fisher Scientific) were homogenized by using zirconium oxide beads and Minilys (Bertin Technologies). The amounts of cytokines and

chemokines in ear homogenate were measured by BD Cytometric Bead Array Flex Set (BD Biosciences, catalog#: 560157, 560232) and LEGENDplex Mouse Proinflammatory Chemokine Panel (BioLegend, catalog#: 740451), according to the manufacturer's protocol.

### In vivo phagocytosis assay

For uptake assay for apoptotic neutrophils, splenic neutrophils were first isolated from the spleen of BALB/c wild-type mice by magnetic sorting using biotinylated anti-Ly6G antibody (clone: 1A8; catalog#: 127604; BioLegend) and Mojosort streptavidin nanobeads (BioLegend, catalog#: 480016). Apoptosis of $Ly6G^+$ neutrophils was then induced by incubating neutrophils for 48 h in RPMI complete medium, followed by labeling with AcidiFluor orange-NHS (Goryo chemical). $1 \times 10^6$ cells of apoptotic neutrophils labeled or unlabeled with AcidiFluor were administered to IgE-CAI skin lesion on day 5 post-challenge. For uptake assay for OVA, 100 μg of OVA labeled or unlabeled with AcidiFluor dye (Goryo chemical) were administered to IgE-CAI skin lesion on day 5 post-challenge. Ear skins were subjected to flow cytometric analysis 2 h after administration of apoptotic neutrophils or OVA.

### scRNA-seq analysis

For single-cell RNA-seq analysis, excised ears were cut into small pieces by using razors, and incubated with Liberase TM (0.25 mg/mL; Roche) in RPMI complete medium at 37 °C for 50 min. After the removal of dead cells and red blood cells through density separation with 25% and 65% Percoll PLUS (Cytiva), isolated cells were incubated with TotalSeq anti-mouse Hashtag-A antibodies [BioLegend, clone: M1/42, 30-F11, dilution 1:400; Hashtag antibody A0311 (catalog #: 155821), A0312 (catalog #: 155823), A0313 (catalog#: 155825) and A0314 (catalog #: 155827)]. Ten thousand labeled cells (approximately 2,500 cells for each condition) were trapped and reverse-transcribed using BD Rhapsody (BD) according to the manufacturer's instructions, followed by the preparation of cDNA libraries and hashtag libraries by TAS-Seq method[50]. The sequencing analysis was conducted by ImmunoGeneTeqs, Inc by using Novaseq 6000 sequencer (Illumina) and Novaseq S4 200 cycles v1.5 kit (Illumina, catalog #: 20028313).

### Data processing for scRNA-seq analysis

After adapter removal and quality filtering by Cutadapt-2.10[51], gene expression libraries were aligned to mouse Ensemble RNA (GRCm38.p6, release-101) by Bowtie2-2.3.4.1[52], and count matrices were generated using the modified Python script of BD Rhapsody workflow. Valid cell barcodes were identified as cell barcodes above the inflection threshold of knee-plot of total read counts of each cell barcode identified by the DropletUtils package[53]. Each sample origin and doublets were identified based on fold-change of the normalized read counts of Hashtag antibodies. To subtract the background read counts of each gene caused by RNA diffusion during the lysis step within the BD Rhapsody cartridge and reverse transcription, distribution-based error correction was performed[50]. The resultant dataset was mainly analyzed using R software package Seurat v4.0.4[54] in R 4.1.0. As quality control, doublets and cells with the mitochondrial gene proportion >20% were filtered out. The log-normalized gene counts were calculated using NormalizeData function (scale.factor = 1,000,000) and highly variable genes were defined by FindVariableFeatures function (selection.method = "vst", nfeature=2000). Read counts were regressed out by the ScaleData function. Principal component analysis was performed on the variable genes, and principal components with their $p$ value < 0.05 calculated by the jackstraw method were subjected to cell clustering and UMAP dimensional reduction. Differentially expressed genes were defined as those whose $p$ value, as calculated by the Wilcoxon rank sum test and adjusted by the Bonferroni method is <0.05 and whose log2FoldChange is >0.5 or

<−0.5. GSEA was conducted by utilizing the R software package clusterProfiler v4.0.5[55]. Pseudotime analysis was performed by utilizing slingshot v2.4.0[56] and tradeSeq[57]. RNA velocity analysis was conducted by utilizing scVelo[58].

### Bulk RNA-seq analysis

For bulk RNA-seq analysis, IgE-CAI skin lesions were cut into small pieces by razors, followed by tissue dissociation by using Multi Tissue Dissociation Kit 1 (Miltenyi Biotec) and gentleMACS Octo Dissociator with Heaters (Miltenyi Biotec) according to the manufacturer's program named 37C_WSDK_2. Cells were then subjected to FACS sorting and four Mo-Mac subpopulations ($Ly6C^{hi}PD-L2^{lo}$, $Ly6C^{hi}PD-L2^{hi}$, $Ly6C^{lo}PD-L2^{hi}$ and $Ly6C^{lo}PD-L2^{lo}$) were isolated. Total RNA was lysed from 5000–80,000 cells sort-purified monocyte-macrophage subpopulations by using lithium dodecyl sulfate-based lysis/storage buffer[59,60]. mRNA was isolated from total RNA by Dynabeads M-270 streptavidin with biotin-oligo (dT)25 (Thermo Fisher Scientific). First- and second-strand cDNA synthesis was performed by Superscript reverse transcriptase IV (Thermo Fisher Scientific) and Kapa HiFi DNA polymerase (Roche). Whole-transcriptome library was subjected to fragmentation/end-repair/A-tailing using NEBNext Ultra II FS DNA Library Prep Kit for Illumina (New England Biolabs, catalog #: E7805). Ligated products were purified and performed the barcoding PCR. Pooled libraries were sequenced by Illumina Novaseq 6000 sequencer (Illumina) and NovaSeq 6000 S4 Reagent Kit v1.5 (Illumina, catalog #: 20028313). Adapter trimming and quality filtering of sequencing data were performed by using Cutadpat-v4.1[51]. The filtered reads were mapped to reference RNA (GRCm38 release-101) using Bowtie2-2.4.5[52], and read number of each gene was counted. Normalization of count data and DE analyses were performed by utilizing the R software package TCC v.1.32.0[61]. PCA visualization was conducted by ggbiplot. Differentially expressed genes (DEGs) for each Mo-Mac subpopulation was defined as follows: DEGs for $Ly6C^+PD-L2^-$ Mo-Mac were defined as genes significantly upregulated in $Ly6C^{hi}PD-L2^{lo}$ Mo-Mac compared to $Ly6C^{hi}PD-L2^{hi}$ and $Ly6C^{lo}PD-L2^{hi}$ Mo-Mac; DEGs for $Ly6C^{hi}PD-L2^{hi}$ Mo-Mac were defined as genes significantly upregulated in $Ly6C^{hi}PD-L2^{hi}$ Mo-Mac compared to $Ly6C^{hi}PD-L2^{lo}$ and $Ly6C^{lo}PD-L2^{hi}$ Mo-Mac; DEGs for $Ly6C^{lo}PD-L2^{hi}$ Mo-Mac were defined as genes significantly upregulated in $Ly6C^{lo}PD-L2^{hi}$ Mo-Mac compared to $Ly6C^{hi}PD-L2^{hi}$ and $Ly6C^{lo}PD-L2^{lo}$ Mo-Mac; DEGs for $Ly6C^{lo}PD-L2^{lo}$ Mo-Mac were defined as genes significantly upregulated in $Ly6C^{lo}PD-L2^{lo}$ Mo-Mac compared to $Ly6C^{lo}PD-L2^{hi}$ and $Ly6C^{hi}PD-L2^{lo}$ Mo-Mac. Module scores for each Mo-Mac subpopulation was calculated by AddModuleScore function in Seurat v4.0.4.

### Quantification and statistical analysis

Statistical analyses were performed with the GraphPad Prism (GraphPad Software). A $p$ value of less than 0.05 was considered statistically significant. Comparisons between the two groups were performed using an unpaired Student's $t$ test. Comparisons between multiple treatment groups and a control group were performed using one-way or two-way ANOVA with post hoc Tukey's multiple comparison test or Sidak's multiple comparison test.

### Reporting summary

Further information on research design is available in the Nature Portfolio Reporting Summary linked to this article.

## Data availability

The scRNA-seq and bulk RNA-seq data generated in this study have been deposited in the NCBI Gene Expression Omnibus (GEO) database under accession codes GSE221310 and GSE245865, respectively. For mapping of transcriptomic data, mouse EnsemblRNA (GRCm38, release-101; http://aug2020.archive.ensembl.org/Mus_musculus/Info/Index) was used. Source data are provided in this paper.

## Code availability

The code is available at GitHub (https://github.com/KensukeMiyake/CMDM-paper, https://doi.org/10.5281/zenodo.10571022).

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

## Acknowledgements

We thank W.A. Kuziel (External Scientific Affairs, Daiichi Sankyo Group, Edison, NJ), N. Mukaida and C. Takahashi (Kanazawa University) for providing *Ccr2*$^{-/-}$ mice, M. Kubo and Y. Suzuki (RIKEN IMS) for providing *Il4ra*$^{fl}$ mice, ImmunoGeneTeqs, Inc. for substantial support for scRNA-seq analysis; all the members of the Karasuyama laboratory for helpful discussions. Cell sorting by using FACS AriaIII and scRNA-seq analysis by using BD Rhapsody system was performed in part at the Research Core of Tokyo Medical and Dental University (TMDU). This work was supported by research grants from the Japanese Ministry of Education, Culture, Sports, Science and Technology [22K007115 (K.M.), 19H01025 (HK), 22H02845 (H.K.)], Takeda Science Foundation (K.M.), KANAE Foundation for the Promotion of Medical Science (K.M.), The Uehara Memorial Foundation (K.M.), The Naito Foundation (K.M.), Ohyama Health Foundation (K.M.), JST SPRING, Grant Number JPMJSP2120 (J.I. and K.T.), Grant-in-Aid for JSPS Fellows 23KJ0837 (J.I.), JST ACT-X, Grant Number JPMJAX232I (K.M.), and the Japan Agency for Medical Research and Development PRIME program (Grant Number JP21gm6210025) (S.S.).

## Author contributions

KM, JI, JN, SS, SY, HK designed the research. KM performed most of the experiments and analyzed data. KM, JI, JN and SS analyzed transcriptome data. KM and HK supervised the work. KM and HK wrote the manuscript. KM, JI, JN, FB, SS, SY, SM and HK provided critical review of the manuscript.

## Competing interests

S.S. reports advisory role for ImmunoGeneTeqs, Inc; stock for ImmunoGeneTeqs, Inc. The other authors have no competing financial interests.
