## [Peer Review File · Nature Communications]

Single-cell transcriptomics identifies the differentiation trajectory from inflammatory monocytes to pro-resolving macrophages in a mouse skin allergy modelREVIEWER COMMENTS

Reviewer #1 (Remarks to the Author):

Authors performed scRNA-seq analysis of IgE-CAI skin lesion and identified four distinct clusters of monocyte/macrophage related populations. They notably identified Ly6ChiPD-L2lo classical monocytes to sequentially differentiate into Ly6CloPD-L2hi macrophages via intermediate Ly6ChiPD-L2hi macrophages but not from Ly6Clo non classical monocytes. Mechanistically, they show that IL-4 receptor-mediated signaling plays an important role in this process and that such macrophages display anti-inflammatory signatures followed by metabolic rewiring concordant with their ability of efferocytosis. Lack of such macrophages (in the *Ccr2*^{-/-} mice) lead to the accumulation of secondary necrotic neutrophils that aggravates skin inflammation. Mechanistically, IL-1 α -stimulated fibroblasts produce neutrophil-attracting chemokines, promoting further recruitment of neutrophils to the IgE-CAI skin lesion of *Ccr2*^{-/-} mice. This is well performed study with interesting results. However, further analysis and experiments need to be performed to support the conclusions made here. Please see below some comments.

On “Thus, the combination of scRNA-seq and flow cytometric analyses enabled us to illustrate that Ly6Chi classical monocytes differentiate into Ly6CloPD-L2hi macrophage via intermediate Ly6ChiPD-L2hi macrophages but not Ly6Clo non-classical monocytes.”, while interesting data, they remain correlative. The authors need to cross validate these data by doing bulk RNA-seq of their identified populations and project back on scRNA-seq data.

Same for “Ly6ChiPD-L2lo classical monocytes isolated from WT mice were incubated ex vivo with culture supernatants of bone marrow-derived basophils (BMBAs) that had been stimulated with IgE plus corresponding allergens (TNP-OVA) (Fig. 2g).”, the authors should compare how such stimulated Ly6ChiPD-L2lo classical monocytes compared to their in vivo data. In addition, performing a similar experiment as in Figure 2G with IL-4 would be nice to identify IL-4 as an important target of mono-mac transition

On “These results suggested that the failure in the generation of highly phagocytic CMDMs in *Ccr2*^{-/-} mice resulted in decreased clearance of apoptotic neutrophils while increased accumulation of necrotic neutrophils, leading to the exaggerated skin inflammation.”, the authors should perform a “rescue” experiment using adoptive transfer of WT monocytes? And show less presence of abscess-like leukocyte aggregates in the IgE-CAI skin lesion of *Ccr2*^{-/-} mice but not of WT mice and other readouts described in figure 4?

On “IL-1 α -stimulated fibroblasts produce neutrophil-attracting chemokines, promoting further recruitment of neutrophils to the IgE-CAI skin lesion of *Ccr2*^{-/-} mice”, how monocytes and fibroblasts interact in WT conditions?

Others:

Gene expression for DCs in Fig. S1D could be expanded for more precise annotation.

Is it possible the early CMDMs must phagocytose in order to become late CMDMs? The lipid metabolic profiling suggests many genes could be a response to breaking down phagocytosed cells.

Figure 4 shows an unsurprising observation. To me this seems like a self-fulfilling prophecy that is not so novel.

Significant handwaving is used to achieve a mechanism in figure 6 of the paper. It is disjointed significantly from the observations in previous figures.

Reviewer #2 (Remarks to the Author):

In the manuscript "Single cell transcriptomics identifies the differentiation trajectory from inflammatory monocytes to pro-resolving macrophages in skin allergies" by Miyake et al. the authors intend to demonstrate that Ly6Chi classical monocytes differentiate into Ly6Clo pro-resolving macrophages through an intermediate stage of monocyte-derived macrophages but without going through non-classical monocytes in inflamed skin. The authors also intend to demonstrate that this process requires IL-4 and that, if it does not occur, necrotic neutrophils accumulate and continue to recruit neutrophils thanks to the release of IL-1a. The authors use a model of allergic skin and single cell RNAseq analyses to show the trajectories leading to pro-resolving macrophages from classical monocytes. The work is interesting and well performed, nevertheless there are some issues that need to be clarified. The work is divided in two parts that are not necessarily correlated. The first part describes the differentiation of pro-resolving macrophages from classical monocytes and the second part describes the role of pro-resolving macrophages in the allergic skin. The second part does not necessarily support the focus of the work that is the differentiation of pro-resolving macrophages from classical monocytes and not from Ly6clo monocytes.

Figure 2d: additional markers, like CD64, should be used to differentiate resident like macrophages indicated in cluster 4 from monocytes (<http://dx.doi.org/10.1016/j.immuni.2013.10.004>).

Figure 2f: in the adoptive transfer experiment the potential conversion of Ly6Chi monocyte to Ly6clo monocytes should be excluded, otherwise the authors cannot be absolutely sure that classical monocytes can generate pro-resolving macrophages.

Figure 2g: the requirement of IL-4 for the differentiation of classical monocytes to pro-resolving macrophages should be demonstrated also in vivo

Figure 3b: how are resident macrophages identified?

Figure 3i: a statistical analysis should be performed

Figure 6: It is not demonstrated that IL-1a is released by dying neutrophils, therefore the hypothesis that in the absence of generation of pro-resolving macrophages necrotic neutrophils themselves maintain their

recruitment is not directly demonstrated

Reviewer #3 (Remarks to the Author):

In the study by Nakayama et al, they investigate the differentiation trajectory from classical inflammatory monocytes to anti-inflammatory macrophages within the skin using a type 1 allergy mouse model. The authors use a combination of single-cell RNA seq, flow cytometry, histology, and functional assays to investigate this. The study is interesting, but several issues need to be clarified before suited for publications:

- 1)Figure 3d – please include the profile for resident macrophages.
- 2)Figure 4a,b,e,f and g: the pictures are very small and some are unclear – please enlarge these.
- 3)Figure 5b – it is surprising that blocking of IL-1beta does not influence the response. Please show the protein level of IL-1alfa and IL-1beta in the model. These findings should also be discussed more in the discussion section.
- 4)Figure 6f. Please add the protein level as well – sometimes one can detect mRNA levels of cytokines/chemokine in fibroblast without proteins being produced.
- 5)The authors use two different protocols to purify cells for Flow and scRNAseq – please explain the reason for this. And how do these protocols impact the phenotype of the cells?
- 6)It is not clear to me how many cells the authors run on their scRNAseq – in the Material session it says 10.000 cells, but is this for each animal or each experiment or in total? If it is in total, it seems like few cells to make all the analyses presented in the study.

Point-by-point responses
Response to Reviewer 1

“Authors performed scRNA-seq analysis of IgE-CAI skin lesion and identified four distinct clusters of monocyte/macrophage related populations. They notably identified Ly6C^{hi}PD-L2^{lo} classical monocytes to sequentially differentiate into Ly6C^{lo}PD-L2^{hi} macrophages via intermediate Ly6C^{hi}PD-L2^{hi} macrophages but not from Ly6C^{lo} non classical monocytes. Mechanistically, they show that IL-4 receptor-mediated signaling plays an important role in this process and that such macrophages display anti-inflammatory signatures followed by metabolic rewiring concordant with their ability of efferocytosis. Lack of such macrophages (in the Ccr2^{-/-} mice) lead to the accumulation of secondary necrotic neutrophils that aggravates skin inflammation. Mechanistically, IL-1 α -stimulated fibroblasts produce neutrophil-attracting chemokines, promoting further recruitment of neutrophils to the IgE-CAI skin lesion of Ccr2^{-/-} mice. This is well performed study with interesting results. However, further analysis and experiments need to be performed to support the conclusions made here. Please see below some comments.”

We greatly appreciate the reviewer’s favorable comments and valuable suggestions to improve our manuscript. According to the Reviewer’s suggestions, we performed additional experiments and revised the manuscript.

Comment #1

“On “Thus, the combination of scRNA-seq and flow cytometric analyses enabled us to illustrate that Ly6C^{hi} classical monocytes differentiate into Ly6C^{lo}PD-L2^{hi} macrophage via intermediate Ly6C^{hi}PD-L2^{hi} macrophages but not Ly6C^{lo} non-classical monocytes.”, while interesting data, they remain correlative. The authors need to cross validate these data by doing bulk RNA-seq of their identified populations and project back on scRNA-seq data.”

We thank the Reviewer for bringing this important issue to our attention. According to the reviewer’s suggestions, we conducted bulk RNA-seq analysis of four monocyte-macrophage subpopulations, namely Ly6C^{hi}PD-L2^{lo}, Ly6C^{hi}PD-L2^{hi}, Ly6C^{lo}PD-L2^{hi}, Ly6C^{lo}PD-L2^{lo} Mo-Macs isolated from IgE-CAI skin lesions on day 5 post-challenge. We conducted the scoring of scRNA-seq data shown in Figure 1f by using differentially expressed genes based on bulk RNA-seq data (see Supplementary Fig. 4 in the revised Ms.). Among four Mo-Mac clusters, cells in classical monocyte cluster (cluster 1) showed the highest scores for Ly6C^{hi}PD-L2^{lo} Mo-Macs, while cells in early CMDM cluster (cluster 2) showed the highest scores for Ly6C^{hi}PD-L2^{hi} Mo-Macs. Cells in late CMDM cluster (cluster 3) showed the highest scores for Ly6C^{lo}PD-L2^{hi} Mo-Macs, while cells in resident-like macrophage cluster (cluster 4) showed the highest scores for Ly6C^{lo}PD-L2^{lo} Mo-Macs. These data indicated that Ly6C^{hi}PD-L2^{lo}, Ly6C^{hi}PD-L2^{hi}, Ly6C^{lo}PD-L2^{hi}, Ly6C^{lo}PD-L2^{lo} Mo-Macs identified by cell surface expression profiles likely corresponded to the classical monocytes, early CMDMs, late CMDMs, and resident-like macrophages identified in scRNA-seq data, respectively. Accordingly, we added bulk RNA-seq data in Supplementary Fig. 4 and revised the Results section of the revised Ms. as follows:

(Results: PP.6-7, Line 131-141 in the revised Ms.)

To further validate the correspondence between each subpopulation of Mo-Macs defined based on the cell surface marker expression and each cluster defined based on the gene expression profile, we conducted bulk RNA-seq analysis of four fractions of Mo-Mac subpopulations (**Supplementary Fig. 4a**). Scoring of scRNA-seq data by differentially expressing genes in each Mo-Mac population revealed that cells in clusters 1, 2, 3 and 4 indeed displayed the highest scores for Ly6C^{hi}PD-L2^{lo}, Ly6C^{hi}PD-L2^{hi}, Ly6C^{lo}PD-L2^{hi} and Ly6C^{lo}PD-L2^{lo} Mo-Macs, respectively (**Supplementary Fig. 4b**). Thus, Ly6C^{hi}PD-L2^{lo}, Ly6C^{hi}PD-L2^{hi}, Ly6C^{lo}PD-L2^{hi} and Ly6C^{lo}PD-L2^{lo} Mo-Macs identified by the cell surface expression profile most likely correspond to classical monocytes, early CMDMs, late CMDMs and resident-like macrophages defined on the basis of scRNA-seq data, respectively.

Comment #2

Same for “Ly6C^{hi}PD-L2^{lo} classical monocytes isolated from WT mice were incubated ex vivo with culture supernatants of bone marrow-derived basophils (BMBAs) that had been stimulated with IgE plus corresponding allergens (TNP-OVA) (Fig. 2g).”, the authors should compare how such stimulated Ly6C^{hi}PD-L2^{lo} classical monocytes compared to their in vivo data.

We thank the Reviewer for bringing this important issue to our attention. According to the reviewer’s suggestions, we conducted additional quantitative PCR analysis for classical monocytes incubated *ex vivo* with supernatants of BMBAs. scRNA-seq analysis of IgE-CAI skin lesion identified that the expression of *Pdcd1lg2* (encoding PD-L2) and *Arg1* (encoding Arginase-1) was rarely detected in classical monocytes, upregulated in early CMDMs, and then gradually downregulated thereafter (Fig. 2c and Figure a below). On the other hand, the expression of *Gas6* was continuously upregulated along the differentiation of Mo-Mac lineage cells (Figure a below). Quantitative PCR analysis revealed that the expression of *Pdcd1lg2* and *Arg1* was significantly upregulated after 24h-incubation with BMBA supernatants, and their expression was unchanged or only slightly upregulated after 48h-incubation (Figure b). On the other hand, *Gas6* expression was significantly upregulated after 48h-incubation (Figure b).

Figure for Reviewer1-1. (a) Ly6C^{hi} classical monocytes isolated from the bone marrow of WT mice were incubated *ex vivo* with culture supernatants of bone marrow-derived basophils (BMBA) that had been stimulated with anti-TNP IgE and TNP-OVA for 18 hr. The mRNA expression of indicated genes before the culture (0h) or after 24hr- or 48hr-incubation with BMBA supernatants are shown (n=3, mean±SEM). The value of mRNA expression in 0h was set as 1. **(b)** Mo-Mac clusters from WT mice shown in Figure 1h were subjected to slingshot pseudotime trajectory analysis. Gene expression changes of indicated genes along with pseudotime ordering are shown.

We therefore concluded that classical monocytes stimulated with BMBA supernatants display similar gene expression profiles to early and late CMDMs *in vivo*, in terms of the expression of *Pdcd1lg2*, *Arg1* and *Gas6*.

“In addition, performing a similar experiment as in Figure 2G with IL-4 would be nice to identify IL-4 as an important target of mono-mac transition.”

We thank the Reviewer for bringing this important issue to our attention. According to the reviewer’s suggestions, we conducted the incubation of Ly6C^{hi} classical monocytes in the presence of recombinant mouse IL-4 (see Supplementary Fig. 6b in the revised Ms.). The 48h-incubation with IL-4 alone promoted the surface phenotype transition into Ly6C^{hi}PD-L2^{hi} early CMDMs and Ly6C^{lo}PD-L2^{hi} late CMDMs, even though the ability of generating late CMDMs appeared lower as compared to BMBA supernatants. To further validate the importance of basophil-derived IL-4, we also conducted the experiment using anti-IL-4 neutralizing antibody as shown in Supplementary Fig. 6a. Accordingly, we added Supplementary Figure 6a-b and revised Ms. as follows:

(Results: P.8 Line 178-187 in the revised Ms.)

Importantly, IL-4 receptor-deficiency in Ly6C^{hi} classical monocytes impaired the change of their surface phenotype unlike WT monocytes when incubated with culture supernatants of allergen TNP-OVA-stimulated basophils (**Fig. 2g**, lower panels). This was also the case when IL-4 neutralizing antibody was added to the culture of WT monocytes (**Supplementary Fig. 6a**). Of note, the treatment of Ly6C^{hi} classical monocytes with recombinant IL-4 promoted the surface phenotype transition into Ly6C^{hi}PD-L2^{hi} and further Ly6C^{lo}PD-L2^{hi}, even though the ability of driving surface phenotype change appeared lower than that of culture supernatants of activated basophils (**Supplementary Fig. 6b**). Thus, the basophil-derived IL-4-IL-4 receptor axis appeared to contribute to the generation of CMDMs from classical monocytes.

Comment #3

On “These results suggested that the failure in the generation of highly phagocytic CMDMs in *Ccr2*^{-/-} mice resulted in decreased clearance of apoptotic neutrophils while increased accumulation of necrotic neutrophils, leading to the exaggerated skin inflammation.”, the authors should perform a “rescue” experiment using adoptive transfer of WT monocytes? And show less presence of abscess-like leukocyte aggregates in the IgE-CAI skin lesion of *Ccr2*^{-/-} mice but not of WT mice and other readouts described in figure 4?

We thank the Reviewer for bringing this important issue to our attention. As shown in Fig. 1d-e, the transfer of WT monocytes into *Ccr2*^{-/-} mice reduced the ear swelling and neutrophil infiltration into the skin lesion. Moreover, the transfer of WT monocytes into *Ccr2*^{-/-} mice attenuated the formation of neutrophil-rich abscess-like leukocyte aggregates (see Supplementary Fig. 11). Accordingly, we revised the Results section of manuscript as follows:

(Results: P.10 Line 255-259 in the revised Ms.)

Adoptive transfer of WT monocytes into *Ccr2*^{-/-} mice attenuated the formation of neutrophil-rich aggregates of leukocytes in the IgE-CAI skin lesion of *Ccr2*^{-/-} mice, suggesting that monocyte-derived macrophages prevented the excessive neutrophil accumulation (Supplementary Fig. 11).

Comment #4

On “IL-1 α -stimulated fibroblasts produce neutrophil-attracting chemokines, promoting further recruitment of neutrophils to the IgE-CAI skin lesion of *Ccr2*^{-/-} mice”, how monocytes and fibroblasts interact in WT conditions?

We thank the Reviewer for bringing this issue to our attention. As shown in supplementary Fig. 13b in the revised Ms., scRNA-seq analysis identified that neutrophils predominantly expressed *Il1a* in the IgE-CAI skin lesion of *Ccr2*^{-/-} mice, suggesting that neutrophils are the one possible source of IL-1 α in IgE-CAI skin lesion. Therefore, it can be assumed that IL-1 α produced by secondary necrotic neutrophils promotes the production of neutrophil-attracting chemokines from fibroblasts, leading to the further recruitment of neutrophils. In WT mice, highly efferocytic CMDMs generated from classical monocytes efficiently remove apoptotic neutrophils to prevent the excess recruitment of neutrophils to the skin lesion. Hence, we assume that monocyte-derived macrophages indirectly suppress IL-1 α -mediated activation of fibroblasts via efferocytic clearance of apoptotic neutrophils. Indeed, CellChat interactome analysis of scRNA-seq data inferred that the interaction strength between Mo-Macs and fibroblast was lower than that of other cell types present in the IgE-CAI skin lesion (as shown in Figure below).

Figure for Reviewer1-2. scRNA-seq datasets of IgE-CAI skin lesion from WT mice shown in Figure 1 was subjected to CellChat interactome analysis (Jin et al. *Nat Commun* 2021). Cell-cell communication networks towards fibroblast are depicted in circle plot, where circle size indicates the number of cells in each cell group and line width represents the communication probability. Mo-Mac: monocyte and monocyte-derived macrophage; Mac: resident-like macrophage; SMC: smooth muscle cell.

To clearly indicate the roles of monocytes in WT mice, we revised the Discussion section of manuscript as follows:

(Discussion: P.15 Line 390-391 in the revised Ms.)

In WT mice, highly efferocytic CMDMs efficiently clear apoptotic cells, thus preventing IL-1 α -mediated aggravation of neutrophilic inflammation.

Minor comment #1

Gene expression for DCs in Fig. S1D could be expanded for more precise annotation.

We thank the Reviewer for bringing this issue to our attention. According to reviewer's suggestions, we further characterized DC subsets shown in Figure S1d by using additional markers. Accordingly, we added additional panels to Supplementary Fig. S1d and revised the Results section of manuscript as follows:

(Results: P.5 Line 91-94 in the revised Ms.)

...while other 2 clusters corresponded to *H2-AbI^{hi}Ciita^{hi}H2-DMb1^{hi}Ly6c1^{int}Ccr2^{hi}Csf1r^{hi}* monocyte-derived DCs (cluster 5) and *H2-AbI^{hi}Ciita^{hi}H2-DMb1^{hi}Flt3^{hi}Cd209a^{hi}Tmem123^{hi}* migratory DCs (cluster 6)²¹ (**Supplementary Fig. 1d**).

Minor comment #2

Is it possible the early CMDMs must phagocytose in order to become late CMDMs? The lipid metabolic profiling suggests many genes could be a response to breaking down phagocytosed cells.

We thank the Reviewer for pointing out this important issue. It has been reported that the efferocytosis of apoptotic cells promotes the differentiation into remodeling macrophages in cooperation with IL-4 and IL-13 (Bosurgi et al. *Science* 2017). Therefore, it is highly probable that the efferocytosis of apoptotic neutrophils promotes the differentiation into late CMDMs. To examine this assumption, we conducted the culture of classical monocytes in the presence of apoptotic neutrophils. The 48h-incubation with apoptotic neutrophils promoted the differentiation from Ly6C^{hi}PD-L2^{lo} classical monocytes into Ly6C^{hi}PD-L2^{hi} early CMDMs and a small fraction of Ly6C^{lo}PD-L2^{hi} late CMDMs (Supplementary Figure 15), suggesting that the engulfment of apoptotic neutrophils could be one driving force for differentiating into early and late CMDMs. Accordingly, we added supplementary Fig. 15 and revised the Discussion section of Ms. as follows:

(Discussion: PP.14-15 Line 366-372 in the revised Ms.)

It has been reported that the efferocytosis of apoptotic cells promotes the differentiation of monocytes to remodeling macrophages in cooperation with IL-4 and IL-13³⁶. Indeed, the 24h- and 48h-incubation of Ly6C^{hi}PD-L2^{lo} classical monocytes *ex vivo* with apoptotic neutrophils induced the change in the surface phenotype to Ly6C^{hi}PD-L2^{hi} and Ly6C^{lo}PD-L2^{lo}, respectively (**Supplementary Fig. 15**), suggesting that the engulfment of apoptotic neutrophils may also promote the differentiation into early and late CMDMs.

Minor comment #3

Figure 4 shows an unsurprising observation. To me this seems like a self-fulfilling prophecy that is not so novel.

We thank the Reviewer for pointing out this important issue. In the first part (Figure 1-3) of our study, we identified the differentiation pathway of Ly6C^{lo} pro-resolving macrophages that possess high efferocytic capacity. In Figure 4 we aimed to identify how the impaired efferocytosis aggravates skin allergic inflammation by using CCR2-deficient mice. Even though the anti-inflammatory effect of efferocytosis is increasingly recognized in various inflammatory contexts (Doran et al. *Nat Rev Immunol* 2020), it remains unclear how the impaired efferocytosis impacts on allergic inflammation.

In the model of allergic airway inflammation several recent reports showed the significance of efferocytosis. Felton et al. showed that impaired efferocytosis by MERTK-deficiency results in increased apoptotic eosinophils, leading to aggravated eosinophilic inflammation (Felton et al. *J Allergy Clin Immunol* 2018). Conversely, a recent report showed that enhanced efferocytosis by progranulin-deficiency alleviates airway allergic inflammation (Huang et al. *Immun Inflamm Dis* 2023). Nevertheless, it remains elusive how the impaired efferocytosis aggravates tissue inflammation especially in the context of skin allergic inflammation.

In Figures 4-6, we addressed the cellular and molecular mechanisms how CCR2-deficiency aggravates skin allergic inflammation. We have identified that IL-1 α but not IL-1 β possibly produced by secondary necrotic neutrophils play critical roles in the neutrophilic aggravation of IgE-CAI in *Ccr2*^{-/-} mice, which is a novel mechanism for aggravation of allergic inflammation by impaired efferocytosis as far as we know. To clearly indicate the purpose of the series of experiments in the second part, we revised Ms. as follows:

(Results: P.10 Line 247-251 in the revised Ms.)

MoMac populations in the IgE-CAI skin lesion of *Ccr2*^{-/-} mice displayed impaired ability of phagocytosing apoptotic neutrophils and allergens (**Fig. 3i**) in accordance with the failure in the generation of CMDMs (**Fig. 1h**). This prompted us to investigate the functional consequence of the failure in the generation of CMDMs *in vivo* by using *Ccr2*^{-/-} mice.

Minor comment #4

Significant handwaving is used to achieve a mechanism in figure 6 of the paper. It is disjointed significantly from the observations in previous figures.

We thank the Reviewer for pointing out this important issue. According to the Reviewer's suggestion, we removed Figure 6 from the main text. Alternatively, we added supplementary Fig. 15 and revised the Discussion section of the Ms. as follows:

(Discussion: P.14 Line 361-367 in the revised Ms.)

The scRNA-seq analysis suggested that skin *Pdgfra*⁺ fibroblasts predominantly expressed the receptor for IL-1a and neutrophil-attracting chemokine CXCL5 (**Supplementary Fig. 15a**). The treatment of *Ccr2*^{-/-} mice with IL-1a neutralizing antibody attenuated the expression of *Cxcl5* in skin fibroblasts (**Supplementary Fig. 15b**), suggesting that skin fibroblasts could be the target of IL-1a which in turn produces CXCL5 to promote neutrophil infiltration into the IgE-CAI skin lesion..

Response to Reviewer 2

“In the manuscript “Single cell transcriptomics identifies the differentiation trajectory from inflammatory monocytes to pro-resolving macrophages in skin allergies” by Miyake et al. the authors intend to demonstrate that Ly6C^{hi} classical monocytes differentiate into Ly6C^{lo} pro-resolving macrophages through an intermediate stage of monocyte-derived macrophages but without going through non-classical monocytes in inflamed skin. The authors also intend to demonstrate that this process require IL-4 and that, if it does not occur, necrotic neutrophils accumulate and continue to recruit neutrophils thank to the release of IL-1α. The authors use a model of allergic skin and single cell RNAseq analyses to show the trajectories leading to pro-resolving macrophages from classical monocytes. The work is interesting and well performed, nevertheless there are some issues that need to be clarified.”

We greatly appreciate the reviewer’s favorable comments and valuable suggestions to improve our manuscript. According to the Reviewer’s suggestions, we performed additional experiments and revised the manuscript.

“The work is divided in two parts that are not necessarily correlated. The first part describes the differentiation of pro-resolving macrophages form classical monocytes and the second part describes the role of pro-resolving macrophages in the allergic skin. The second part does not necessarily support the focus of the work that is the differentiation of pro-resolving macrophages from classical monocytes and not from Ly6c^{lo} monocytes.”

We thank the Reviewer for pointing out this important issue. In the first part of our study, we identified that the Ly6C^{lo} pro-resolving macrophages were differentiated from Ly6C^{hi} classical monocytes but not Ly6C^{lo} non-classical macrophages. During this analysis, we revealed that Ly6C^{hi} pro-resolving macrophages possess high efferocytic capacity. Based on these results, we addressed the functional roles of pro-resolving macrophages by using CCR2-deficient mice in the second part. We have identified in the second part of the study that the impaired efferocytosis in *Ccr2*^{-/-} mice promotes the increase of IL-1α levels in the skin lesion, leading to the neutrophilic aggravation of IgE-CAI.

In the first part of the study, we have also revealed that IL-4 receptor signaling plays key roles in the differentiation from classical monocytes into early and late CMDMs. Indeed, IL-4 receptor-deficient mice impaired the generation of early and late CMDMs in the IgE-CAI skin lesion (Figure a below). As in the case with *Ccr2*^{-/-} mice, *Il4ra*^{-/-} mice showed neutrophilic aggravation of IgE-CAI which is ameliorated by the administration of anti-IL-1α neutralizing antibody (Figure b-c below).

Figure for Reviewer 2. *Il4ra*^{-/-} and WT mice were treated as in Fig. 1 to induce IgE-CAI. (a) Cells isolated from the IgE-CAI skin lesion on day 5 were subjected to flow cytometric analysis. The surface expression of PD-L2 and Ly6C in Mo-Mac populations were shown. (b-c) Neutralizing antibody against IL-1 α or its isotype-matched control antibody (HmIgG) was intraperitoneally administered to indicated mice on days 0, 1, 2, 3, and 4 post-challenge. In b, time course of ear swelling (Δ ear thickness) is shown (mean \pm SEM, n=2-3 each). In c, the number of hematopoietic cells and neutrophils isolated from the ear skin of indicated mice on day 5 is shown (mean \pm SEM, n=2-3 each).

Therefore, the second part of the study in part supports the functional aspects of the roles of pro-resolving macrophages identified in first part of the study. We believe that showing both the functional significance and differentiation trajectory of pro-resolving macrophages would be of great of importance for elucidating the mechanisms for the resolution of allergic inflammation. To bridge the gap between the first and the second part of this study, we clearly indicated the purpose of the series of experiments in the beginning of the second part of Results section in the revised Ms. as follows:

(Results: P.10 Line 247-251 in the revised Ms.)

MoMac populations in the IgE-CAI skin lesion of *Ccr2*^{-/-} mice displayed impaired ability of phagocytosing apoptotic neutrophils and allergens (**Fig. 3i**) in accordance with the failure in the generation of CMDMs (**Fig. 1h**). This prompted us to investigate the functional consequence of the failure in the generation of CMDMs *in vivo* by using *Ccr2*^{-/-} mice.

Moreover, we removed the description on Figure 6 from the main text. Alternatively, we added supplementary Fig. 15 and revised the Discussion section of the Ms. as follows:

(Discussion: P.14 Line 361-367 in the revised Ms.)

The scRNA-seq analysis suggested that skin *Pdgfra*⁺ fibroblasts predominantly expressed the receptor for IL-1a and neutrophil-attracting chemokine CXCL5 (**Supplementary Fig. 15a**). The treatment of *Ccr2*^{-/-} mice with IL-1a neutralizing antibody attenuated the expression of *Cxcl5* in skin fibroblasts (**Supplementary Fig. 15b**), suggesting that skin fibroblasts could be the target of IL-1a which in turn produces CXCL5 to promote neutrophil infiltration into the IgE-CAI skin lesion.

Comment#1

“Figure 2d: additional markers, like CD64, should be used to differentiate resident like macrophages indicated in cluster 4 from monocytes (<http://dx.doi.org/10.1016/j.immuni.2013.10.004>).”

We thank the Reviewer for bringing this important issue to our attention. According to the reviewer’s suggestions, we included additional markers to characterize Ly6C^{lo}PD-L2^{lo} Mo-Mac population. As shown in supplementary Figure 3a in the revised Ms., the majority (~80%) of Ly6C^{lo}PD-L2^{lo} Mo-Mac were CD64⁺ macrophages. Accordingly, we added supplementary Fig. S3a-b and revised the Results section of the Ms. as follows:

(Results: P.6 Line 129-131 in the revised Ms.)

In accordance with this categorization, the majority of the latter 3 populations expressed macrophage markers²², CD64 (FcγRI) and F4/80 on their cell surface (**Supplementary Fig. 3a-b**).

Comment #2

“Figure 2f: in the adoptive transfer experiment the potential conversion of Ly6C^{hi} monocyte to Ly6C^{lo} monocytes should be excluded, otherwise the authors cannot be absolutely sure that classical monocytes can generate pro-resolving macrophages.”

We thank the Reviewer for bringing this important issue to our attention. To exclude the possibility that Ly6C^{hi}PD-L2^{hi} late CMDMs were generated through Ly6C^{lo} non-classical monocytes, we conducted the adoptive transfer of CD45.1⁺Ly6C^{lo} non-classical monocytes (see Supplementary Fig. 5 in the revised Ms.). Unlike Ly6C^{hi} classical monocytes, the majority of adoptively transferred Ly6C^{lo} non-classical monocytes remained Ly6C^{lo}PD-L2^{lo} on day 5. Even though 10% of transferred Ly6C^{lo} non-classical monocytes showed Ly6C^{lo}PD-L2^{int} phenotype, they showed little or no expression of CD64 (macrophage marker) on their cell surface. Therefore, we concluded that it seems highly unlikely that pro-resolving Ly6C^{hi}PD-L2^{hi} late CMDMs were differentiated from Ly6C^{lo} non-classical monocytes. Accordingly, we added Supplementary Fig. 5 and revised the Results section of the Ms. as follows:

(Results: P.7 Line 157-160 in the revised Ms.)

On the other hand, when Ly6C^{lo} non-classical monocytes were adoptively transferred into CD45.2⁺ mice, the majority of them remained Ly6C^{lo}PD-L2^{lo}CD64^{-/lo}F4/80^{-/lo} even on day 5 post-challenge (**Supplementary Fig. 5**).

Comment #3

“Figure 2g: the requirement of IL-4 for the differentiation of classical monocytes to pro-resolving macrophages should be demonstrated also in vivo.”

We thank the Reviewer for bringing this important issue to our attention. According to the reviewer’s suggestions, we assessed the Mo-Mac populations in Mo-Mac-specific IL-4 receptor-deficient mice (*Cx3cr1*^{Cre/+} *Il4ra*^{fl} mice) (see Supplementary Fig. 6c in the revised Ms).

Mo-Mac specific-IL-4 receptor-deficiency reduced the frequency of both Ly6C^{hi}PD-L2^{hi} early CMDMs and Ly6C^{lo}PD-L2^{hi} late CMDMs, indicating a significant role of IL-4 receptor-mediated signaling in the differentiation of early and late CMDMs. Accordingly, we added Supplementary Fig. 6c and revised the manuscript as follows:

(Results: P.8 Line 187-191 in the revised Ms.)

To verify these observations *in vivo*, we established macrophage-specific IL-4-receptor deficient mice (*Cx3cr1*^{Cre/+} *Il4ra*^{fl} mice). Macrophage-specific IL-4 receptor deficiency impaired the transition of classical monocytes into early and late CMDMs (**Supplementary Fig. 6c**), suggesting the involvement of IL-4 receptor-mediated signaling in the transition *in vivo*.

Comment #4

“Figure 3b: how are resident macrophages identified?”

We thank the Reviewer for pointing out this issue. We added gating markers (Ly6C^{lo}PD-L2^{lo-int}) for resident macrophages in Figure 3b in the revised Ms.

Comment #5

“Figure 3i: a statistical analysis should be performed.”

We thank the Reviewer for pointing out this issue. We added graphs with statistics to Fig. 3i, showing the significantly reduced uptake of apoptotic neutrophils and OVA in Mo-Macs from CCR2-deficient mice.

Comment #6

“Figure 6: It is not demonstrated that Il-1a is released by dying neutrophils, therefore the hypothesis that in the absence of generation of pro-resolving macrophages necrotic neutrophils itself maintain their recruitment is not directly demonstrated.”

We thank the Reviewer for bringing this important issue to our attention. According to the reviewer’s suggestions, we assessed which cell types in the IgE-CAI skin lesion mainly expressed *Il1a*. scRNA-seq analysis identified that neutrophils predominantly expressed *Il1a* in the IgE-CAI skin lesion of *Ccr2*^{-/-} mice, suggesting that neutrophils are the one possible source of IL-1α in IgE-CAI skin lesion. Accordingly, we added Supplementary Fig. 13b and revised the manuscript as follows:

(Results: P.11 Line 281-285 in the revised Ms.)

Providing that the *Il1a* mRNA expression was predominantly expressed in neutrophils in *Ccr2*^{-/-} mice (**Supplementary Fig. 13b**), it can be assumed that IL-1α produced by inflammatory cells, most likely dying neutrophils, promotes the accumulation of neutrophils in the IgE-CAI skin lesion of *Ccr2*^{-/-} mice.

Response to Reviewer 3

“In the study by Nakayama et al, they investigate the differentiation trajectory from classical inflammatory monocytes to anti-inflammatory macrophages within the skin using a type 1 allergy mouse model. The authors use a combination of single-cell RNA seq, flow cytometry, histology, and functional assays to investigate this. The study is interesting, but several issues need to be clarified before suited for publications.”

We greatly appreciate the reviewer’s favorable comments and valuable suggestions to improve our manuscript. According to the Reviewer’s suggestions, we performed additional experiments and revised the manuscript.

Comment #1

“Figure 3d – please include the profile for resident macrophages.”

We thank the Reviewer for bringing this issue to our attention. According to the reviewer’s suggestions, we added the gene expression profile for resident-like macrophages to Figure 3d. Even though resident-like macrophages displayed high expression of efferocytic receptors and their bridging molecules, phagocytosis associated genes tend to be enriched in late CMDMs as compared to resident-like macrophages (Supplementary Fig. 10). Moreover, genes associated with PI3K-Akt signaling pathway, one key signaling pathway of efferocytic receptors, was significantly enriched in late CMDMs, compared to resident-like macrophages (supplementary Fig. 10). This might be why resident-like macrophages showed lower phagocytic ability compared to late CMDMs (Figure 3e-h). Accordingly, we added supplementary Fig. 10 and revised the Results section of the Ms. as follows:

(Results: P.10 Line 234-243 in the revised Ms.)

Of note, resident-like macrophages showed reduced phagocytic ability, particularly in the phagocytosis of apoptotic neutrophils, compared to late CMDMs (Fig. 3 e-h) even though they displayed high expression of genes encoding efferocytic receptors and the bridging molecules (Fig. 3d). Considering that the expression of genes associated with phagocytosis tended to decline in resident-like macrophages compared to late CMDMs (Supplementary Fig. 10a), the expression of those associated with the process following the recognition of apoptotic cells might decline in resident-like macrophages. Indeed, the expression of genes associated with PI3K-Akt signaling, one key signaling pathway downstream of efferocytic receptors, declined in resident-like macrophages compared to late CMDMs (Supplementary Fig. 10b).

Comment #2

“Figure 4a,b,e,f and g: the pictures are very small and some are unclear – please enlarge these.”

We thank the Reviewer for bringing this important issue to our attention. According to the reviewer’s suggestions, we enlarged HE-stained images shown in Figures 4a, b, e, f and g in the revised Ms.

Comment #3

“Figure 5b – it is surprising that blocking of IL-1beta does not influence the response. Please show the protein level of IL-1alfa and IL-1beta in the model. These findings should also be discussed more in the discussion section.”

We thank the Reviewer for bringing this important issue to our attention. According to the Reviewer’s suggestion, we measured IL-1 α and IL-1 β proteins in the IgE-CAI skin lesion of WT and CCR2-deficient mice (see Supplementary Fig. 13a). The amounts of IL-1 α protein were approximately 60 times higher than those of IL-1 β . Moreover, CCR2-deficient mice showed higher IL-1 α amounts in the skin lesion compared to WT mice. Accordingly, we revised the manuscript as follows:

(Results: P. 11 Line 278-281 in the revised Ms.)

In line with this observation, the amounts of IL-1 α protein in the IgE-CAI skin lesion were significantly higher in *Ccr2*^{-/-} mice than in WT mice. Of note, the amounts of IL-1 β were ~60 times lower than those of IL-1 α (Supplementary Fig. 13a).

Comment #4.

“Figure 6f. Please add the protein level as well – sometimes one can detect mRNA levels of cytokines/chemokine in fibroblast without proteins being produced.”

We thank the Reviewer for bringing this important issue to our attention. According to the Reviewer’s suggestions, we measured CXCL5 protein in the skin lesion of WT and *Ccr2*^{-/-} mice. The amounts of CXCL5 protein were significantly higher in *Ccr2*^{-/-} mice than in WT mice. (Figure a below). Considering that the expression of *Cxcl5* was predominantly detected in fibroblasts (Figure b below), it is likely that fibroblasts are the major producers of CXCL5 protein in the skin lesion of *Ccr2*^{-/-} mice.

Figure for Reviewer3-1. (a) WT and *Ccr2*^{-/-} mice were treated as in Fig. 1 to induce IgE-CAI. The amount of CXCL5 in tissue homogenates collected from the IgE-CAI skin lesions on day 5 is shown (n=3 mice each, mean \pm SEM) (b) scRNA-seq datasets in Fig. 1f were analyzed. Feature plots showing the expression of *Cxcl5* in the IgE-CAI skin lesion of *Ccr2*^{-/-} mice are shown.

Comment #5.

“The authors use two different protocols to purify cells for Flow and scRNAseq – please explain the reason for this. And how do these protocols impact the phenotype of the cells?”

We thank the Reviewer for bringing this important issue to our attention. Before conducting scRNA-seq analysis we compared skin cell dissociation methods for flow-cytometric and scRNA-seq analysis. For flow-cytometric analysis, we dissociated skin tissue by collagenase, whereas for scRNA-seq analysis we dissociated skin tissue by Liberase enzyme. As shown in Figure below, Liberase-based dissociation yielded a greater number of skin cells and higher frequency of live cells. On the other hand, Liberase-based dissociation significantly reduced the cell surface expression of some proteins such as PD-L2, possibly due to the shedding of cell surface proteins by Liberase. Therefore, we used collagenase-based method for flowcytometric analysis, but we used Liberase-based dissociation for scRNA-seq analysis.

Figure for Reviewer3-2. BALB/c WT mice were treated as in Figure 1 to induce IgE-CAI. On day 5 post-challenge, the IgE-CAI skin lesion was dissociated with collagenase (Flow-cytometry protocol) or Liberase (scRNA-seq protocol) and subjected to flow cytometric analysis. PI-FSC^{int-hi} live cells were gated in the left panels. The surface expression of Ly6C and PD-L2 is displayed in the right panels.

Comment #6.

“It is not clear to me how many cells the authors run on their scRNAseq – in the Material session it says 10.000 cells, but is this for each animal or each experiment or in total? If it is in total, it seems like few cells to make all the analyses presented in the study.”

We thank the Reviewer for bringing this important issue to our attention. Ten thousand cells in total (approximately 2,500 cells for each condition) were analyzed by scRNA-seq. Since we utilized TAS-seq method (the recently developed high-resolution scRNA-seq analysis), we can analyze in-depth gene expression analysis even in low input of the cell number According to the Reviewer’s suggestion, we revised the Methods section of manuscript as follows:

(Methods: P.21 Line 558-560 in the revised Ms.)

Ten thousand labeled cells (**approximately 2,500 cells for each condition**) were trapped and reverse-transcribed using BD Rhapsody (BD) according to the manufacturer's instructions,...

REVIEWERS' COMMENTS

Reviewer #1 (Remarks to the Author):

The authors have adequately revised their manuscript and improve their study. Congratulations!

Reviewer #2 (Remarks to the Author):

The authors satisfactorily answered all my concerns. I do not have any additional requests.

Reviewer #3 (Remarks to the Author):

In their revision, the authors improved their manuscript and also responded adequately to all my comments.